# LABEL DISTRIBUTION LEARNING VIA IMPLICIT DISTRIBUTION REPRESENTATION

## ABSTRACT

In contrast to multi-label learning, label distribution learning characterizes the polysemy of examples by a label distribution to represent richer semantics. In the learning process of label distribution, the training data is collected mainly by manual annotation or label enhancement algorithms to generate label distribution. Unfortunately, the complexity of the manual annotation task or the inaccuracy of the label enhancement algorithm leads to noise and uncertainty in the label distribution training set. To alleviate this problem, we introduce the implicit distribution in the label distribution learning framework to characterize the uncertainty of each label value. Specifically, we use deep implicit representation learning to construct a label distribution matrix with Gaussian prior constraints, where each row component corresponds to the distribution estimate of each label value, and this row component is constrained by a prior Gaussian distribution to moderate the noise and uncertainty interference of the label distribution dataset. Finally, each row component of the label distribution matrix is transformed into a standard label distribution form by using the self-attention algorithm. In addition, some approaches with regularization characteristics are conducted in the training phase to improve the performance of the model.

## 1 INTRODUCTION

Label distribution learning (LDL) ( Geng (2016)) is a novel learning paradigm that characterizes the polysemy of examples. In LDL, the relevance of each label to an example is given by an exact numerical value between 0 and 1 (also known as description degree), and the description degree of all labels forms a distribution to fully characterize the polysemy of an example. Compared with traditional learning paradigms, LDL is a more generalizable and representational learning paradigm that provides richer semantic information.

LDL has been successful in several application domains (Gao et al. (2018); Zhao et al. (2021); Chen et al. (2021a); Si et al. (2022)). To obtain the label distribution for learning, there are mainly two ways: one is expert labeling, but labeling is expensive and there is no objective labeling criterion, and the resulting label distribution is highly subjective and ambiguous. The other is to convert a multi-label dataset into a label distribution dataset through a label enhancement algorithm (Xu et al. (2019; 2020); Zheng et al. (2021a); Zhao et al. (2022b)). However, label enhancement lacks a reliable theory to ensure that the label distribution recovered from logical labels converges to the true label distribution, because logical labels provide a very loose solution space for the label distribution, making the solution less stable and less accurate.

In summary, the label distribution dataset used for training has a high probability of inaccuracy and uncertainty, which significantly limits the performance of LDL algorithms. To characterize and mitigate the uncertainty of the label distribution, we propose a novel LDL method based on the implicit label distribution representation. Our work is inspired by recent work on implicit neural representation in 2D image reconstruction (Sitzmann et al. (2020)). The key idea of implicit neural representation is to represent an object as a function that maps a sequence of coordinates to the corresponding signal, where the function is de-parameterized by a deep neural network. In this paper, we start with a deep network to extract the latent features of input information. Then, the latent features are looked up against the encoded coordinate matrix to generate a label distribution matrix (implicit distribution representation). Finally, the label distribution matrix is computed by a self-attention module to yield a standard label distribution. Note that the goal of the proposed implicit distribution representation is to generate a label distribution matrix with Gaussian distribution constraints as a customized representation pattern.

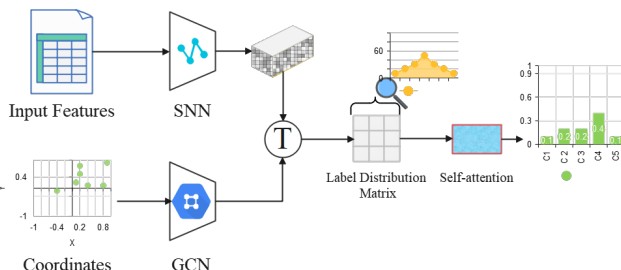

Figure 1: **Our architecture.** This figure shows the architecture of the proposed deep implicit function, which consists of two parts. The first part starts with a latent feature prediction stream (SNN with an MLP) that learns the input information to predict the feature maps. The second part learns a label distribution matrix to regress a label distribution.

To efficiently generate latent features around the coordinate, we design a deep spiking neural network (Yamazaki et al. (2022)) with an MLP as an executor to extract latent features in the input information. The architecture of the whole network consists of multiple layers of linear spiking neurons, and the neurons of different layers conduct a shortcut between them. Notably, spiking neural networks have two key properties that are different from the representation of artificial neural networks. First, a standard spiking neural network considers the time characteristics $T$ (taking the image as an example, the input tensor $X \in \mathbb{R}^{T \times C \times W \times H}$.) at a multi-step inference mechanism. Here, we create several pseudo-feature spaces on the native feature space by setting different strategies with data augmentation (Ucar et al. (2021)). These pseudo-features and the native feature are stacked in the time dimension $T$ to achieve the multi-step inference mechanism. Second, the representation capability of the spiking neural network is underpowered, since the output space is a binarized sequence ($\{0 \cdots 1\}$). Therefore, we place a standard MLP in the last layer of the spiking neural network, which projects the features into the real number space. Our model saves about **30∼40**% of energy consumption over ANNs with the same network structure on embedded devices such as lynxi HP300, Raspberry Pi, or Apple smartphones (PyTorch 1.2 support M1 Mac).

The extraction of latent features provides material for the coordinates to generate the label distribution matrix. First, the initialized coordinate matrix (the size is $L \times 64$, where $L$ denotes the number of labels, and 64 denotes features of the nodes) is reconstructed by using a GCN. Note that the features of the nodes meet the Gaussian distribution since the deep network needs to reconstruct the data from a fixed distribution. Then the coordinate matrix is repeated in $N$ copies, and $N$ denotes the number of samples. Next, the coordinate matrix computes a label distribution matrix (the size is $N \times L \times 2L$) in the latent feature space by looking up the table[1]. Each component of the label distribution matrix represents the distribution of each label value, and the components are constrained by a priori Gaussian distributions. Finally, the label distribution matrix leverages a self-attention mechanism to obtain the corresponding label distribution of the samples.

Yet, deep learning-based approaches are prone to overfitting manually extracted features. To alleviate the problem, we propose some regularization approaches to boost the performance of the model, and a new dataset based on the image comprehension task is released. **Our contribution includes: (i)** For LDL, this is a novel method to obtain the label distribution of a sample through the implicit distribution representation. **(ii)** Spiking neural network with an MLP is developed to save energy consumption of mobile devices, and correlations between labels are deeply mined by a graph convolutional network. **(iii)** To the best of our knowledge, we are the first to tackle the tabular LDL issue by using deep learning. Facing the LDL task, some regularization techniques are designed to boost the performance of the model and a new LDL dataset is released.

## 2 BACKGROUND AND MOTIVATION

Starting in 2016, LDL ( Geng (2016)) is officially proposed as a novel learning paradigm that aims to inscribe the polysemy of a sample through the degree of description. Then, from the viewpoint of task kinds, LDL is categorized into two domains. **1)** Addressing the uncertainty of application tasks ( Gao et al. (2017); Ren & Geng (2017); Gao et al. (2018); Chen et al. (2021a); Liu et al. (2021); Zhao et al. (2021); Li et al. (2022); Si et al. (2022); Cao et al. (2022); Buisson et al. (2022)); **2)** Studying the characteristics of label distributions on customized datasets ( Geng (2016); Zhao & Zhou (2018); Ren et al. (2019b;a); Wang & Geng (2021); Jia et al. (2021a;b); Zhao et al. (2022a); Tan et al. (2022)). However, the existing work overlooks the fact that task **2** also requires uncertainty modeling for the label space. In this paper, we conduct uncertainty modeling on task **2** to boost the learning ability of the LDL algorithm. From a technical viewpoint, our approach has three

---

[1] https://pytorch.org/tutorials/intermediate/spatial_transformer_tutorial.html

distinct motivations for uncertainty estimation. (i) To avoid the limitations of explicit modeling on the regression space, we introduce deep learning to handle task **2**, where uncertainty is enforced on the representation process (label distribution matrix). By modeling the label space, the label distribution matrix provides a smoother estimation space as a feature map in the deep network. (ii) Compared to existing work ( Ren et al. (2019b); Jia et al. (2021a); Qian et al. (2022)) that considers label correlation, we use GCN to deeply mine label correlation. (iii) Besides that, SNN started to show amazing potential on regression tasks ( Ahmadi et al. (2021); Patel et al. (2021); Kim & Panda (2021); Lian et al. (2022); Yamazaki et al. (2022)), to enhance computational efficiency and maintain modeling capabilities, thus introducing it to handle task **2**. Overall, our approach tackles a new problem, while the technical approach is the first of its kind.

## 3 PROPOSED METHOD

As shown in Figure 1, our architecture is a two-stage scheme, which is based on an implicit representation approach. In the first stage (**latent feature extraction**), the raw features $\mathcal{X}$ are fed into the encoder for encoding to construct a latent feature space $F_f \in \mathbb{R}^{L \times H \times W}$. In the second stage (**label distribution matrix learning**), the coordinate tensor (coordinate matrix is reshaped after GCN) $C_r \in \mathbb{R}^{L \times L \times 2)}$ is looked up in the latent feature space $F_f$ to generate a label distribution matrix $\mathcal{M} \in \mathbb{R}^{L \times 2L}$, and finally a label distribution $\mathcal{D}$ is obtained by using self-attention algorithm. The dimension 2 of the coordinate tensor comes from a weak assumption that the label relation exists in a two-dimensional plane, such as the relative positions of "tree" and "sun" in an image. Furthermore, some regularization terms are introduced to our model to boost its performance.

**Latent Feature Extraction.** Our model starts with a latent feature extraction task, which is based on a spiking neural network. Spiking neural networks have a high potential value as a 3rd deep model and are efficient and interpretable. In the LDL task, we develop a non-fixed Network Architecture and a Network Implementation due to the diversity of LDL dataset formats.

**Network Architecture:** Our network consists of 17 layers of function units, which include 8 linear units, 8 nonlinear units, and a transformation layer (including an MLP, a mean operator, and a reshape operator). Essentially, this is a residual network and the transformation layer at the tail of the network. Except for the first layer (the number of neurons is the dimensionality of the input features), each linear unit contains 1024 neurons, followed closely by a ReLU non-linearity, and the last layer is a reshape function to generate the feature space $F_f$. On datasets with a smaller feature space $\mathcal{X}$ (e.g., Gene dataset has only 36 features), each linear layer includes 64 neurons. We also attempted to use other activation functions than ReLU, such as PReLU, Swish, and Sigmoid, but without any advantage. Besides, since the spiking network involves a time dimension, the time dimension $T$ is squeezed in the output space of the spiking network by using a mean operator. The network's tailgate should notice that the feature map is reshaped as $F_f$ using a reshape operator and the MLP (the number of neurons is $L \times W \times H$.), where $H$ and $W$ are 32.

**Network Implementations:** The simplified residual network is implemented as an ANN in two frameworks: PyTorch 1.12 (a standard deep network training library) and SpikingJelly 0.0.13 (a library for deep learning applications that is part of the PyTorch ecosystem). The SpikingJelly implementation can be run in spiking or non-spiking modes. To train the spiking network, we use an ANN-SNN conversion training approach. The conversion is simple, we convert the trained model on PyTorch with the help of an ann2snn.Converter (SpikingJelly) to obtain an SNN model. Note that the conversion method defines six modes (max, $99.9\%$, $1.0\backslash2$, $1.0\backslash3$, $1.0\backslash4$, $1.0\backslash5$) to obtain SNN with different accuracy, and in this paper, we chose $99.9\%$. The literature ( Patel et al. (2021)) provides a theoretical basis for analyzing the conversion of ANN to SNN.

**Learning Label Distribution Matrix.** Inspired by confident learning (Northcutt et al. (2021)), we develop a label distribution matrix with a Gaussian before estimating the uncertainty of the labels. For the data distribution of each label value, we use a Gaussian distribution to delimit the distribution rather than other multi-peaked distribution priors, and numerous kinds of literature have verified that this approach can eliminate the uncertainty (Liu et al. (2021); Zheng et al. (2021b); Ghosh et al. (2021); Li et al. (2022)). Note that since the label distribution values are in the range of 0 to 1, we also constrain the vector after Gaussian sampling to be in the range of 0 to 1. Furthermore, to capture the global correlation between labels to generate a standard label distribution, we employ a self-attention mechanism to model the label distribution matrix. To obtain a label distribution matrix $\mathcal{M}$, we need a latent feature $F_f$ from the SNN and a coordinate matrix $C_b : \text{GCN}(C_r)$ (GCN denotes the graph convolution network) that passes through the graph convolution network. Specifically, first, we initialize a matrix of coordinates $C_r \in \mathbb{R}^{L \times 64}$ based on the functions (torch.randn) provided by

PyTorch. $L$ denotes the number of nodes and $64$ denotes that we assign one feature vector to each node, with each feature vector sampled in a Gaussian distribution. To build a graph structure the data needs information about the edges, where there is an edge with no direction between the nodes. So far, we built the node with the edge information and have the ability to integrate it into a graph to input into the GCN. Our GCN includes four graph convolution layers and four activation layers, where the activation function uses ReLU. These four graph convolutions include 64, 128, 256, and $L \times L \times 2$ neurons respectively. There is one key message to note, the output matrix $C_b$ of the GCN is filled (torch.repeat) with the same number of samples as the latent feature space $F_f$. Then, we use $C_b$ to look up the table in $F_f$ to obtain a matrix by using flatten operations. The $L$ label distribution values include a $1 \times 2L$ vector to form a label distribution matrix $\mathcal{M}$. Each vector is constrained by a designated Gaussian distribution $\bar{\mathcal{M}}$ with parameters whose mean is the value of the label distribution and variance of 0.5. Finally, this matrix is squeezed by using a self-attention algorithm (Vaswani et al. (2017)) to obtain the corresponding label distribution for the samples. For self-attention algorithm, we treat each vector in the label distribution matrix as a token by using the scaled dot product to obtain a global correlation matrix, and finally an MLP with Softmax to squeeze the matrix for a standard label distribution. As shown in Figure 2, we extract the heat map calculated by the self-attention algorithm, heat map of the model output before the self-attention algorithm and the heat map of raw label space on Movie dataset. Overall, the label correlations of these heat maps tend to be consistent.

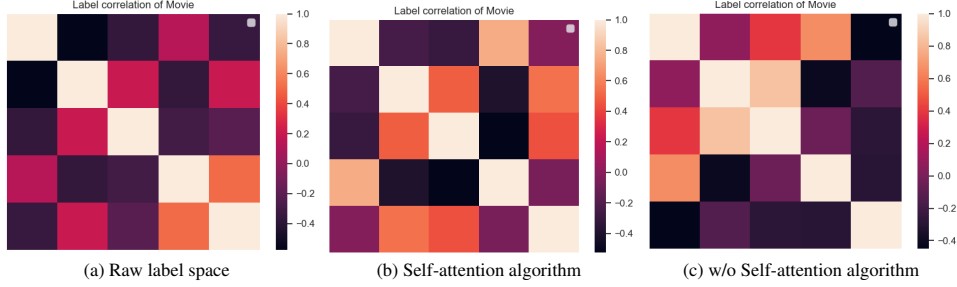

| (a) Raw label space | (b) Self-attention algorithm | (c) w/o Self-attention algorithm |

Figure 2: This figure shows the label correlation on the Movie dataset.

**Regularization Techniques.** We propose two techniques (Linear Normalization Function, Data Augmentation) to boost the performance of our model.

**Linear Normalization Function:** Currently, most existing LDL algorithms use Softmax to output a vector in the tail of the model.

$$\text{Softmax}(d_i^{y_i}) = \frac{e^{d_i^{y_i}}}{\sum_{k=1}^{K} e^{d_k^{y_k}}}, \tag{1}$$

where $d_i^{y_i}$ denotes a label value in the vector and $K$ denotes the number of elements. Since the complexity of exponential operations is much higher than linear operations, especially in the training phase of the model (e.g. Gamma correction (Ju et al. (2019))). Several algorithms (such as cosFormer (Qin et al. (2022))) seek to replace exponential operations. Here, we develop a linear function Lnf to replace Softmax to output a label distribution vector. Then predicted label distribution $\widehat{\mathbf{d}}_i = \left\{ \widehat{d_i^{y_1}}, \widehat{d_i^{y_2}}, \ldots, \widehat{d_i^{y_c}} \right\}$ can be obtained by:

$$\text{Lnf}(\widehat{d_i^{y_i}}) = \frac{d_i^{y_i} + |\mathcal{D}_{\min}|}{\sum_{c=1}^{C}(d_c^{y_c} + |\mathcal{D}_{\min}|)}, \tag{2}$$

where $|\mathcal{D}_{\min}|$ denotes the absolute value of the minimum of the predicted label distribution values. Essentially, our method offsets the values of the label distribution to the positive domain of the x-axis. We evaluate both Softmax and linear normalization function on 12 datasets, and our network with the linear normalization function converges $2.1 \times$ faster than the model with Softmax.

**Data Augmentation**: Most LDL datasets are difficult to augment the samples with expert knowledge since the input features are extracted manually. To overcome this challenge, we introduce a simple and data-agnostic data augmentation routine, termed mixup (Zhang et al. (2018)). In a nutshell, mixup constructs virtual training examples:

$$\begin{aligned} \tilde{x} &= \lambda x_i + (1 - \lambda)x_j, & \text{where } x_i, x_j \text{ are raw input vectors} \\ \tilde{y} &= \lambda \mathbf{d}_i + (1 - \lambda)\mathbf{d}_j, & \text{where } \mathbf{d}_i, \mathbf{d}_j \text{ are label distribution vectors} \end{aligned} \tag{3}$$

$(x_i, y_i)$ and $(x_j, y_j)$ are two examples drawn at random from our training data, and $\lambda \in [0, 1]$. However, a deep network may still be over-fitted during the training phase, which leads to poor generalization. We use the random masking scheme endowed on mixup, formally expressed as:

$$
\begin{aligned}
\tilde{x} &= \lambda x_i * \text{mask} + (1 - \lambda)x_j * \text{mask}, && \text{where } x_i, x_j \text{ are raw input vectors} \\
\tilde{y} &= \text{Lnf}(\lambda \mathbf{d}_i + (1 - \lambda)\mathbf{d}_j), && \text{where } \mathbf{d}_i, \mathbf{d}_j \text{ are label distribution vectors}
\end{aligned}
\tag{4}
$$

the mask is a vector represented by 1 or 0. In this paper, a mask contains 80% of the scalar 1 and the rest is 0. Moreover, to solidify the definition of the label distribution, we use Lnf to normalize the synthesized label vectors. We also tried using other regularization schemes, such as random cropping, and the results do not improve significantly.

**Loss Function.** We optimize the weights and biases of the proposed network by minimizing the $L_1$, KullbackLeibler (K-L) divergence, and regularization of the label distribution matrix on the training set,

$$
\mathcal{L} = \frac{1}{N}\sum_{i=1}^{N}\left\|\widehat{\mathbf{d}}_i - \mathbf{d}_i\right\|_1 + \lambda\mathcal{L}_{kl}(\widehat{\mathbf{d}}_i, \mathbf{d}_i) + \beta\sum_{n=1}^{N}\sum_{l=1}^{L}\mathcal{L}_2(\mathcal{M}_{nl}, \bar{\mathcal{M}}_{nl}),
\tag{5}
$$

where $N$ is the number of training samples, $\widehat{\mathbf{d}}_i$ is the label distribution vector by our model, and $\mathbf{d}_i$ is the corresponding ground truth. The weight $\lambda$ of the loss term $\mathcal{L}_{kl}$ (K-L) is set to 0.01 in our experiments. In addition, the weight $\beta$ of the loss term (label distribution matrix) is set to 0.1 in our experiments. Since it is difficult to capture useful information by encoding a small number of tokens, encoding for sequences of long tokens can extract higher-order semantics, such as perceptual loss (Johnson et al. (2016)). Therefor, we used the above-mentioned learning strategy on the datasets with several labels less than 20. On the datasets with the number of labels greater than 20, we used $L_1$, K-L divergence, perceptual loss, and regularization of the label distribution matrix on the training set,

$$
\mathcal{L} = \frac{1}{N}\sum_{i=1}^{N}\left\|\widehat{\mathbf{d}}_i - \mathbf{d}_i\right\|_1 + \lambda_1\mathcal{L}_{kl}(\widehat{\mathbf{d}}_i, \mathbf{d}_i) + \lambda_2\mathcal{L}_p + \beta\sum_{n=1}^{N}\sum_{l=1}^{L}\mathcal{L}_2(\mathcal{M}_{nl}, \bar{\mathcal{M}}_{nl}),
\tag{6}
$$

we use MLPs as the pre-trained model for the perceptual loss function ($\mathcal{L}_p$). The weight $\lambda_1$, $\lambda_2$, and $\beta$ are set to 0.01, 0.01, 0.08 in our experiments, respectively. These MLPs consist of three linear layers with fixed neurons and three activation layers (ReLU), where the number of neurons is the same as the number of labels, using Kaiming initialization (He et al. (2015)).

## 4 EXPERIMENTS

**Algorithm Configurations.** We conduct experiments on 12 datasets, the characteristics of the datasets are reported in Table 5. Except for dataset wc-LDL, the configuration of all other datasets is referenced to (Wang & Geng (2021)). This new release dataset (wc-LDL) has 500 watercolor images and corresponding label distribution (12 emotions). We develop the SNN with configurations on different datasets also summarized in Table 5. To evaluate the performance of LDL models, we use the six metrics proposed by (Geng (2016)), including Chebyshev distance ↓, Clark distance ↓, Canberra distance ↓, KL divergence ↓, Cosine similarity ↑, and Intersection similarity ↑. LF and DA denote the loss function and data augmentation method, respectively. ↓ represents the indicator's performance favoring low values and ↑ represents the indicator's performance favoring high values. $\mathscr{L}_{1,2:end}$ denotes the number of neurons in the head layer and the rest of the layers of the SNN.

**Experimental Setting.** We conduct comparative experiments with five LDL algorithms (BFGS-LLD (Geng (2016)), LDL-LRR (Jia et al. (2021a)), LDL-LCLR (Ren et al. (2019b)), LDLSF (Ren et al. (2019a)) and LALOT (Zhao & Zhou (2018))) which also used data augmentation schemes on 12 datasets. Furthermore, to validate the effectiveness of our proposed method, we design a baseline model to be conducted on 12 datasets. For the baseline model (a-LDL), we drop the implicit representation and use only a two-layer network with the label distribution matrix to learn a label distribution. a-LDL without any additional regularization terms and data augmentation mechanisms in the training stage. BFGS-LLD is based on a linear model, the loss function is K-L divergence, and the optimization method is the quasi-Newton approach. LDL-LRR and LDL-LCLR both consider label correlations in the learning process, with the former considering the order relationship of the labels and the latter capturing global relationships between labels. For LDL-LRR, the parameters $\lambda$ and $\beta$ are selected from $10^{\{-6,-5,...,-2,-1\}}$ and $10^{\{-3,-2,...,1,2\}}$, respectively. For LDL-LCLR, the

parameters $\lambda_1, \lambda_2, \lambda_3, \lambda_4$ and $k$ are set to $0.0001, 0.001, 0.001, 0.001$ and 4, respectively. LDLSF leverages label-specific features and common features simultaneously, whose parameters $\lambda_1, \lambda_2$ and $\lambda_3$ are selected from $10^{\{-6,-5,\ldots,-2,-1\}}$, respectively, and $\rho$ is set to $10^{-3}$. LALOT adopts optimal transport distance as the loss function, the trade-off parameter $C$ and the regularization coefficient $\lambda$ is set to 200 and 0.2, respectively. Our approach for experimental settings is reported in Table 6. It is worth noting that Early stopping and Greed soup (Wortsman et al. (2022)) are also used on all the datasets where the comparison algorithm is executed. Our method is marked as gray .

| Dataset | Algorithm | Chebyshev ↓ | Clark ↓ | Canberra ↓ | K-L ↓ | Cosine ↑ | Intersection ↑ |
|---|---|---|---|---|---|---|---|
| wc-LDL | Ours | 0.0779 ± 0.0021 | 0.3980 ± 0.0051 | 0.7779 ± 0.0030 | 0.4040 ± 0.0020 | 0.9883 ± 0.0009 | 0.8778 ± 0.0014 |
| | a-LDL | 0.0855 ± 0.0034 | 0.4667 ± 0.0052 | 0.8007 ± 0.0110 | 0.4455 ± 0.0033 | 0.9788 ± 0.0079 | 0.8705 ± 0.0079 |
| | LDL-LRR | 0.1122 ± 0.0030 | 0.4772 ± 0.0036 | 0.8802 ± 0.0024 | 0.5533 ± 0.0049 | 0.9510 ± 0.0022 | 0.8555 ± 0.0047 |
| | LDL-LCLR | 0.1057 ± 0.0019 | 1.0569 ± 0.0039 | 0.7890 ± 0.0039 | 0.5045 ± 0.0037 | 0.9668 ± 0.0049 | 0.8383 ± 0.0018 |
| | LDLSF | 0.1009 ± 0.0038 | 0.4199 ± 0.0044 | 0.9008 ± 0.0015 | 0.5199 ± 0.0040 | 0.9779 ± 0.0018 | 0.8660 ± 0.0022 |
| | LALOT | 0.0989 ± 0.0019 | 0.6689 ± 0.0019 | 0.8089 ± 0.0049 | 0.4778 ± 0.0018 | 0.9476 ± 0.0020 | 0.8700 ± 0.0033 |
| | BFGS-LLD | 0.1229 ± 0.0039 | 1.5657 ± 0.0021 | 0.7998 ± 0.0020 | 0.4998 ± 0.0051 | 0.9704 ± 0.0036 | 0.8611 ± 0.0016 |
| SJAFFE | Ours | 0.0854 ± 0.0018 | 0.4008 ± 0.0030 | 0.7955 ± 0.0023 | 0.4100 ± 0.0012 | 0.9799 ± 0.0014 | 0.8809 ± 0.0015 |
| | a-LDL | 0.0899 ± 0.0112 | 0.4189 ± 0.0123 | 0.8019 ± 0.0023 | 0.4131 ± 0.0012 | 0.9702 ± 0.0014 | 0.8702 ± 0.0015 |
| | LDL-LRR | 0.1122 ± 0.0030 | 0.4772 ± 0.0036 | 0.8802 ± 0.0024 | 0.5533 ± 0.0049 | 0.9510 ± 0.0022 | 0.8555 ± 0.0047 |
| | LDL-LCLR | 0.1057 ± 0.0019 | 1.0569 ± 0.0039 | 0.7890 ± 0.0039 | 0.5045 ± 0.0037 | 0.9668 ± 0.0049 | 0.8383 ± 0.0018 |
| | LDLSF | 0.1123 ± 0.0038 | 0.4397 ± 0.0044 | 0.9212 ± 0.0015 | 0.5557 ± 0.0040 | 0.9779 ± 0.0018 | 0.8660 ± 0.0022 |
| | LALOT | 0.0989 ± 0.0019 | 0.6689 ± 0.0019 | 0.8089 ± 0.0049 | 0.4778 ± 0.0018 | 0.9476 ± 0.0020 | 0.8700 ± 0.0033 |
| | BFGS-LLD | 0.1334 ± 0.0139 | 1.6648 ± 0.0023 | 0.7999 ± 0.0022 | 0.04778 ± 0.0051 | 0.9711 ± 0.0036 | 0.8655 ± 0.0116 |
| SBU | Ours | 0.0833 ± 0.0020 | 0.3994 ± 0.0010 | 0.7611 ± 0.0020 | 0.3650 ± 0.0014 | 0.9811 ± 0.0015 | 0.8900 ± 0.0017 |
| | a-LDL | 0.0954 ± 0.0041 | 0.4099 ± 0.0010 | 0.7774 ± 0.0083 | 0.4557 ± 0.0014 | 0.9788 ± 0.0041 | 0.8754 ± 0.0010 |
| | LDL-LRR | 0.1109 ± 0.0036 | 0.4477 ± 0.0039 | 0.8666 ± 0.0026 | 0.5344 ± 0.0028 | 0.9597 ± 0.0029 | 0.8592 ± 0.0033 |
| | LDL-LCLR | 0.1100 ± 0.0025 | 0.9660 ± 0.0039 | 0.7897 ± 0.0033 | 0.5101 ± 0.0021 | 0.9677 ± 0.0056 | 0.8555 ± 0.0032 |
| | LDLSF | 0.1117 ± 0.0048 | 0.4199 ± 0.0044 | 0.9013 ± 0.0015 | 0.5199 ± 0.0040 | 0.9780 ± 0.0029 | 0.8622 ± 0.0022 |
| | LALOT | 0.0989 ± 0.0019 | 0.6689 ± 0.0019 | 0.8421 ± 0.0049 | 0.4776 ± 0.00168 | 0.9476 ± 0.0020 | 0.8700 ± 0.0033 |
| | BFGS-LLD | 0.1119 ± 0.0030 | 1.4657 ± 0.0022 | 0.7700 ± 0.0025 | 0.04932 ± 0.0053 | 0.9753 ± 0.0036 | 0.8710 ± 0.0019 |
| Scene | Ours | 0.2998 ± 0.0020 | 2.3374 ± 0.0018 | 6.5163 ± 0.0018 | 0.8111 ± 0.0029 | 0.7890 ± 0.0049 | 0.5691 ± 0.0010 |
| | a-LDL | 0.3111 ± 0.0046 | 2.3881 ± 0.0043 | 6.6189 ± 0.0066 | 0.8111 ± 0.0029 | 0.7516 ± 0.0049 | 0.5600 ± 0.0111 |
| | LDL-LRR | 0.3889 ± 0.0111 | 3.1698 ± 0.0031 | 6.8777 ± 0.0025 | 0.8999 ± 0.0069 | 0.7044 ± 0.0077 | 0.5444 ± 0.0049 |
| | LDL-LCLR | 0.3740 ± 0.0066 | 2.4986 ± 0.0066 | 6.8600 ± 0.0067 | 0.8559 ± 0.0039 | 0.7119 ± 0.0122 | 0.5119 ± 0.0081 |
| | LDLSF | 0.3441 ± 0.0249 | 2.9884 ± 0.0055 | 6.6900 ± 0.0055 | 0.8391 ± 0.0044 | 0.7336 ± 0.0088 | 0.5660 ± 0.0041 |
| | LALOT | 0.3129 ± 0.0152 | 2.3999 ± 0.0044 | 6.6666 ± 0.0078 | 0.8226 ± 0.0033 | 0.7390 ± 0.0100 | 0.5224 ± 0.0066 |
| | BFGS-LLD | 0.3598 ± 0.0020 | 2.4998 ± 0.0033 | 6.7999 ± 0.0049 | 0.8400 ± 0.0033 | 0.7333 ± 0.0064 | 0.5199 ± 0.0055 |
| Gene | Ours | 0.0488 ± 0.0012 | 2.1029 ± 0.0259 | 14.0888 ± 0.0551 | 0.2335 ± 0.0044 | 0.8395 ± 0.0032 | 0.7984 ± 0.0066 |
| | a-LDL | 0.0502 ± 0.0032 | 2.1777 ± 0.0211 | 14.1221 ± 0.0413 | 0.2443 ± 0.0045 | 0.8298 ± 0.0022 | 0.7889 ± 0.0063 |
| | LDL-LRR | 0.0537 ± 0.0039 | 2.2887 ± 0.0860 | 14.3550 ± 0.0144 | 0.2559 ± 0.0077 | 0.8288 ± 0.0144 | 0.7789 ± 0.0040 |
| | LDL-LCLR | 0.0511 ± 0.0022 | 2.2201 ± 0.0444 | 14.2101 ± 0.0510 | 0.2566 ± 0.0047 | 0.8302 ± 0.0012 | 0.7722 ± 0.0060 |
| | LDLSF | 0.0513 ± 0.0030 | 2.2221 ± 0.0036 | 14.3667 ± 0.0265 | 0.2445 ± 0.0077 | 0.8320 ± 0.0010 | 0.7701 ± 0.0026 |
| | LALOT | 0.0505 ± 0.0033 | 2.1989 ± 0.0194 | 14.1855 ± 0.0922 | 0.2443 ± 0.0088 | 0.8297 ± 0.0060 | 0.7888 ± 0.0013 |
| | BFGS-LLD | 0.0578 ± 0.0066 | 2.3008 ± 0.0188 | 14.3559 ± 0.1556 | 0.2480 ± 0.0015 | 0.8300 ± 0.0049 | 0.7786 ± 0.0070 |
| Movie | Ours | 0.1089 ± 0.0018 | 0.5001 ± 0.0044 | 0.9722 ± 0.0040 | 0.0977 ± 0.0008 | 0.9485 ± 0.0061 | 0.8602 ± 0.0006 |
| | a-LDL | 0.1121 ± 0.0023 | 0.5199 ± 0.0098 | 1.0549 ± 0.0043 | 0.1141 ± 0.0031 | 0.9477 ± 0.0063 | 0.8566 ± 0.0046 |
| | LDL-LRR | 0.1135 ± 0.0009 | 0.5244 ± 0.0010 | 1.1551 ± 0.0061 | 0.1445 ± 0.0049 | 0.9510 ± 0.0022 | 0.8772 ± 0.0007 |
| | LDL-LCLR | 0.1177 ± 0.0086 | 0.5345 ± 0.0040 | 1.1533 ± 0.0111 | 0.1559 ± 0.0030 | 0.9360 ± 0.0049 | 0.8222 ± 0.0011 |
| | LDLSF | 0.1155 ± 0.0045 | 0.5339 ± 0.0062 | 1.1152 ± 0.0050 | 0.1540 ± 0.0041 | 0.9445 ± 0.0020 | 0.8551 ± 0.0044 |
| | LALOT | 0.1221 ± 0.0110 | 0.5440 ± 0.0033 | 1.1110 ± 0.0040 | 0.1503 ± 0.0008 | 0.9477 ± 0.0022 | 0.8559 ± 0.0002 |
| | BFGS-LLD | 0.1310 ± 0.0032 | 0.5230 ± 0.0022 | 1.1170 ± 0.0024 | 0.1595 ± 0.0155 | 0.9400 ± 0.0003 | 0.8491 ± 0.0018 |
| M2B | Ours | 0.3763 ± 0.0022 | 1.1560 ± 0.0102 | 2.0889 ± 0.0055 | 0.4880 ± 0.0023 | 0.7998 ± 0.0022 | 0.6703 ± 0.0033 |
| | a-LDL | 0.3810 ± 0.0032 | 1.2998 ± 0.0143 | 2.1222 ± 0.0023 | 0.4889 ± 0.0099 | 0.7801 ± 0.0039 | 0.6610 ± 0.0066 |
| | LDL-LRR | 0.3993 ± 0.0010 | 1.4990 ± 0.0166 | 2.1884 ± 0.0034 | 0.5246 ± 0.0006 | 0.7531 ± 0.0023 | 0.6334 ± 0.0077 |
| | LDL-LCLR | 0.4040 ± 0.0082 | 1.2444 ± 0.0045 | 2.2000 ± 0.0009 | 0.4996 ± 0.0013 | 0.7760 ± 0.0079 | 0.6555 ± 0.0012 |
| | LDLSF | 0.4159 ± 0.0055 | 1.3105 ± 0.0041 | 2.2155 ± 0.0076 | 0.5002 ± 0.0006 | 0.7552 ± 0.0004 | 0.6234 ± 0.0033 |
| | LALOT | 0.3881 ± 0.0099 | 1.4883 ± 0.0012 | 2.1257 ± 0.0268 | 0.4990 ± 0.0008 | 0.7549 ± 0.0021 | 0.6620 ± 0.0053 |
| | BFGS-LLD | 0.3811 ± 0.0044 | 1.3650 ± 0.0002 | 2.1992 ± 0.0095 | 0.4995 ± 0.0005 | 0.7699 ± 0.0040 | 0.6532 ± 0.0009 |
| SCUT | Ours | 0.3895 ± 0.0021 | 1.2140 ± 0.0111 | 2.1995 ± 0.0095 | 0.4911 ± 0.0030 | 0.6990 ± 0.0002 | 0.6504 ± 0.0001 |
| | a-LDL | 0.3992 ± 0.0022 | 1.2149 ± 0.0133 | 2.2002 ± 0.0022 | 0.4990 ± 0.0006 | 0.6800 ± 0.0032 | 0.6466 ± 0.0009 |
| | LDL-LRR | 0.4159 ± 0.0010 | 1.6680 ± 0.0122 | 2.2006 ± 0.0039 | 0.5388 ± 0.0006 | 0.6531 ± 0.0023 | 0.5804 ± 0.0007 |
| | LDL-LCLR | 0.4240 ± 0.0042 | 1.3444 ± 0.0055 | 2.2450 ± 0.0016 | 0.5131 ± 0.0022 | 0.6261 ± 0.0005 | 0.5500 ± 0.0012 |
| | LDLSF | 0.4360 ± 0.0015 | 1.2185 ± 0.0022 | 2.2159 ± 0.0076 | 0.5120 ± 0.0006 | 0.6261 ± 0.0004 | 0.5534 ± 0.0030 |
| | LALOT | 0.3999 ± 0.0009 | 1.4983 ± 0.0012 | 2.2207 ± 0.0158 | 0.4995 ± 0.0002 | 0.6549 ± 0.0020 | 0.6411 ± 0.0044 |
| | BFGS-LLD | 0.3992 ± 0.0055 | 1.5656 ± 0.0163 | 2.2832 ± 0.0080 | 0.4966 ± 0.0011 | 0.6491 ± 0.0040 | 0.6333 ± 0.0013 |
| fbp5500 | Ours | 0.1251 ± 0.0002 | 1.1890 ± 0.0120 | 2.0980 ± 0.0223 | 0.1053 ± 0.0009 | 0.9643 ± 0.0015 | 0.8501 ± 0.0025 |
| | a-LDL | 0.1272 ± 0.0023 | 1.1990 ± 0.0111 | 2.1008 ± 0.0244 | 0.1102 ± 0.0022 | 0.9600 ± 0.0033 | 0.8499 ± 0.0034 |
| | LDL-LRR | 0.1313 ± 0.0031 | 1.2519 ± 0.0038 | 2.1992 ± 0.0095 | 0.1127 ± 0.0077 | 0.9533 ± 0.0021 | 0.8412 ± 0.0066 |
| | LDL-LCLR | 0.1277 ± 0.0016 | 1.1969 ± 0.0039 | 2.1194 ± 0.0046 | 0.1135 ± 0.0006 | 0.9588 ± 0.0044 | 0.8483 ± 0.0014 |
| | LDLSF | 0.1270 ± 0.0028 | 1.1909 ± 0.0164 | 2.1846 ± 0.0119 | 0.1193 ± 0.0041 | 0.9609 ± 0.0019 | 0.8460 ± 0.0007 |
| | LALOT | 0.1306 ± 0.0022 | 1.1921 ± 0.0015 | 2.1111 ± 0.0171 | 0.1120 ± 0.0015 | 0.9430 ± 0.0019 | 0.8400 ± 0.0004 |
| | BFGS-LLD | 0.1299 ± 0.0049 | 1.4655 ± 0.0041 | 2.1675 ± 0.0024 | 0.1135 ± 0.0055 | 0.9595 ± 0.0030 | 0.8419 ± 0.0018 |

Table 1: The performance of our proposed method with the comparison algorithms on 12 datasets.

**Results and Discussion.** We conduct 10 times 5-fold cross-validation on each dataset. The experimental results are presented in the form of "mean±std" in Tables 1 and 7. Overall, our proposed method outperforms other comparison algorithms in all evaluation metrics. Each comparison algo-

rithm employs some regularization techniques to expand the training sample as well as to prevent overfitting, however, four main factors contribute to the competitive results of our approach. **i):** Moderate noise, especially on the Gene dataset, due to the uncertainty that comes with manual annotation, our approach has a huge performance gain with the help of implicit distribution representation with Gaussian priors. **ii):** The ability to capture global features between labels with the help of a self-attention mechanism. Also, a-LDL performs sub-optimally probably because the depth of the model is insufficient. **iii):** The powerful representational capabilities of the deep network, especially on image datasets, give us a huge advantage.

| AS | Algorithm | Chebyshev ↓ | Clark ↓ | Canberra ↓ | K-L ↓ | Cosine ↑ | Intersection ↑ | Dataset |
|---|---|---|---|---|---|---|---|---|
| (a) | Ours | $0.0488 \pm 0.0012$ | $2.1029 \pm 0.0259$ | $14.0888 \pm 0.0551$ | $0.2335 \pm 0.0044$ | $0.8395 \pm 0.0032$ | $0.7984 \pm 0.0066$ | Gene |
| | w/o $\mathcal{L}_p$ | $0.0497 \pm 0.0021$ | $2.1664 \pm 0.0177$ | $14.2651 \pm 0.0155$ | $0.2488 \pm 0.0071$ | $0.8290 \pm 0.074$ | $0.7884 \pm 0.0037$ | |
| (b,c,d) | Ours | $0.0779 \pm 0.0021$ | $0.3980 \pm 0.0051$ | $0.7779 \pm 0.0030$ | $0.04040 \pm 0.0020$ | $0.9883 \pm 0.0009$ | $0.8778 \pm 0.0014$ | wc-LDL |
| | w/o PRT | $0.0877 \pm 0.0009$ | $0.4008 \pm 0.0043$ | $0.7881 \pm 0.0014$ | $0.04223 \pm 0.0010$ | $0.9779 \pm 0.0008$ | $0.8699 \pm 0.0012$ | |
| | SNN | $0.0771 \pm 0.0012$ | $0.4006 \pm 0.0047$ | $0.7805 \pm 0.0011$ | $0.04118 \pm 0.0016$ | $0.9801 \pm 0.0012$ | $0.8664 \pm 0.0029$ | |
| | GNN | $0.0804 \pm 0.0033$ | $0.4133 \pm 0.0017$ | $0.7968 \pm 0.0020$ | $0.04991 \pm 0.0052$ | $0.9705 \pm 0.0036$ | $0.8612 \pm 0.0016$ | |

Table 2: Ablation study (AS). Effectiveness of the loss functions and the modules on two datasets. Quantitative results demonstrate the effectiveness of each module.

**Ablation Study.** To demonstrate the effectiveness of the loss function and the module of our model, we conduct an ablation study involving the following four experiments: **(a)** w/o perceptual loss function $\mathcal{L}_p$: we remove the loss function on training Gene dataset, shown in Table 2. **(b)** w/o our proposed regularization techniques (PRT): we remove the linear normalization function and the data augmentation respectively on the training wc-LDL dataset, shown in Table 2. **(c)** The effectiveness of SNN: we use standard MLPs to replace SNNs in the same network architecture, shown in Table 2. **(d)** The effectiveness of GNN: for deep implicit function construction, we use standard MLPs to replace GNNs, as shown in Table 2. **(e)** mixup with mask vs. mixup: we evaluate these two methods on 12 data sets with our proposed model, and the results show that using mixup with mask improve overall performance by 6% over using mixup. We conduct 10 times 5-fold cross-validation on the dataset of the ablation experiment.

**Potential of Model and Network.** Our model can be extended to handle classification and semi-supervised tasks. We conduct some experiment reports to show the potential of the public datasets. In addition, there is a deep model (DLDL (Gao et al. (2017))) based on a label distribution learning framework as a training scheme being performed on several classification tasks. Since DLDL does not have an adapted network implementation on tabular data, to demonstrate the effectiveness of our proposed method, our approach battles with the DLDL algorithm in the age estimation dataset. (1) Facial expression recognition (Zhao et al. (2021)): To evaluate the effectiveness of the proposed model, we conduct the experiments on the public in-the-wild facial expression datasets (RAF-DB, CAER-S, AffectNet). For images on all the datasets, the face region is detected and aligned by using Retinaface (Deng et al. (2020)). Then, the image is cropped to a fixed resolution ($224 \times 224$) by bi-linear interpolation. Our approach is pre-trained on the face recognition dataset MS-Celeb-1M, and the 50-layer Residual Network is used as the backbone network. For our network, parameters were optimized using the Adam optimizer with an initial learning rate of 0.01, a mini-batch size of 128, and an epoch is 50. Notably, our network architecture uses a model (both input and output layers are changed in the number of neurons and the rest are fixed) executed on the Gene dataset with a single RTX GPU, and the loss function uses cross-entropy. We compared it with the current SOTA model (Face2Exp) on three popular datasets. The recognition accuracy of Face2Exp in the three data sets (RAF-DB, CAER-S, AffectNet) is 88.54%, 86.16%, and 64.23% respectively. The recognition accuracy of our proposed model in three datasets is 88.52%, 86.36%, and 65.02% respectively. Our algorithm achieves competitive results on most of the face recognition datasets. (2) MedM-NIST Classification Decathlon (Yang et al. (2021)): We evaluate the algorithm's performance on the MedMNIST Classification Decathlon benchmark. The area under the ROC curve (AUC) and Accuracy (ACC) is used as the evaluation metrics. Our approach is pre-trained on the Gene dataset, the 18-layer Residual Network is adopted as the backbone network. Our model is trained for 100 epochs, using a cross-entropy loss and an Adam optimizer with a batch size of 128 and an initial learning rate of $1 \times 10^{-3}$ The overall performance of the methods is reported in Table 3. Our model is pre-trained during the training phase and thus achieves competitive performance on most of the datasets. Literature (Yang et al. (2021)) includes the bibliography of the full comparison method. (3) CIFAR-10, SVHN, CIFAR-100 (Hu et al. (2021)): Since our approach involves graph structure, it battles with other SOTAs on the semi-supervised task. We use ResNet-28-2 as our backbone and Adam with

| Methods | PathMNIST | | ChestMNIST | | DermaMNIST | | OCTMNIST | | PneumoniaMNIST | |
|---|---|---|---|---|---|---|---|---|---|---|
| | AUC | ACC | AUC | ACC | AUC | ACC | AUC | ACC | AUC | ACC |
| ResNet-18 (28) | 0.972 | 0.844 | 0.706 | 0.947 | 0.899 | 0.721 | 0.951 | **0.758** | 0.957 | 0.843 |
| ResNet-18 (224) | 0.978 | 0.860 | 0.713 | **0.948** | 0.896 | 0.727 | 0.960 | 0.752 | 0.970 | 0.861 |
| ResNet-50 (28) | 0.979 | **0.864** | 0.692 | 0.947 | 0.886 | 0.710 | 0.939 | 0.745 | 0.949 | 0.857 |
| ResNet-50 (224) | 0.978 | 0.848 | 0.706 | 0.947 | 0.895 | 0.719 | 0.951 | 0.750 | 0.968 | 0.896 |
| auto-sklearn | 0.500 | 0.186 | 0.647 | 0.642 | 0.906 | 0.734 | 0.883 | 0.595 | 0.947 | 0.865 |
| AutoKeras | 0.979 | **0.864** | 0.715 | 0.939 | 0.921 | 0.756 | 0.956 | 0.736 | 0.970 | 0.918 |
| Google AutoML Vision | 0.982 | 0.811 | 0.718 | 0.947 | 0.925 | **0.766** | 0.965 | 0.732 | **0.993** | 0.941 |
| Ours | **0.985** | **0.864** | **0.722** | **0.948** | **0.929** | 0.765 | **0.969** | 0.756 | 0.992 | **0.943** |

| Methods | RetinaMNIST | | BreastMNIST | | OrganMNIST_A | | OrganMNIST_C | | OrganMNIST_S | |
|---|---|---|---|---|---|---|---|---|---|---|
| | AUC | ACC | AUC | ACC | AUC | ACC | AUC | ACC | AUC | ACC |
| ResNet-18 (28) | 0.727 | 0.515 | 0.897 | 0.859 | 0.995 | 0.921 | 0.990 | 0.889 | 0.967 | 0.762 |
| ResNet-18 (224) | 0.721 | 0.543 | 0.915 | 0.878 | **0.997** | 0.931 | 0.991 | 0.907 | **0.974** | 0.777 |
| ResNet-50 (28) | 0.719 | 0.490 | 0.879 | 0.853 | 0.995 | 0.916 | 0.990 | 0.893 | 0.968 | 0.746 |
| ResNet-50 (224) | 0.717 | 0.555 | 0.863 | 0.833 | **0.997** | 0.931 | **0.992** | 0.898 | 0.970 | 0.770 |
| auto-sklearn | 0.694 | 0.525 | 0.848 | 0.808 | 0.797 | 0.563 | 0.898 | 0.676 | 0.855 | 0.601 |
| AutoKeras | 0.655 | 0.420 | 0.833 | 0.801 | 0.996 | 0.929 | **0.992** | 0.915 | 0.972 | 0.803 |
| Google AutoML Vision | **0.762** | 0.530 | **0.932** | 0.865 | 0.988 | 0.818 | 0.986 | 0.861 | 0.964 | 0.706 |
| Ours | 0.760 | **0.570** | 0.931 | **0.890** | 0.996 | **0.933** | 0.990 | **0.917** | 0.971 | **0.810** |

Table 3: **Overall performance of MedMNIST** in metrics of AUC and ACC, using ResNet-18 / ResNet-50 with resolution 28 and 224, auto-sklearn , AutoKeras and Google AutoML Vision.

| Methods | CIFAR-10 | | SVHN | | CIFAR-100 | | Backbone |
|---|---|---|---|---|---|---|---|
| | 1000 labels | 4000 labels | 1000 labels | 4000 labels | 10000 labels | 15000 labels | |
| VAT | 81.36 | 88.95 | 94.02 | 95.80 | 77.54 | 81.55 | WRN-28-8 |
| MeanTeacher | 82.68 | 89.64 | 96.25 | 96.61 | 72.68 | 79.62 | WRN-28-8 |
| MixMatch | 92.25 | 93.76 | 96.73 | 97.11 | 71.69 | 79.13 | WRN-28-8 |
| ReMixMatch | 94.27 | 94.86 | 97.17 | 97.58 | 76.97 | 80.99 | WRN-28-8 |
| FixMatch | - | 95.69 | **97.64** | - | 77.40 | 64.25 | WRN-28-8 |
| SimPLE | 94.84 | 94.95 | 97.54 | 97.31 | 78.11 | **82.40** | WRN-28-8 |
| Ours | **94.98** | **95.99** | 97.56 | **97.62** | **78.64** | 81.02 | WRN-28-8 |

Table 4: CIFAR-10, CIFAR-100, and SVHN Top-1 test accuracy.

weight decay for optimization in all experiments. Our model is trained for 100 epochs, using a cross-entropy loss with a batch size of 128 and an initial learning rate of $1 \times 10^{-2}$ Early-stopping and data augmentation (Mixup) are also adopted in the training phase. Two datasets (CIFAR-10, and SVHN) are used for us to evaluate the performance of the algorithm. The overall performance of the methods is reported in Table 4. Our model still selects MLPs pre-trained on the Gene dataset and performs competitively on the semi-supervised task. Literature (Hu et al. (2021)) includes the bibliography of the full comparison method. (4) Semi-supervised label distribution learning (Jia et al. (2021b)): To evaluate the capability of our model, our proposed algorithm is compared to the SOTA model (PGE-SLDL (Jia et al. (2021b))) in the Gene with a 50% missing rate. We conduct it 10 times on Gene and average the 10 results as the final result. PGE-SLDL yield $0.0059\pm0.0008$, $0.4298\pm0.0006$, $4.3899\pm0.0005$, $0.09788\pm0.0005$, $0.8911\pm0.0008$ and $0.8789\pm0.0007$ respectively on the data set with 6 evaluation metrics (Chebyshev↓, Clark↓, Canberra↓, K-L divergence↓, Intersection↑, Cosine↑). Our method yield $0.0053\pm0.0012$, $0.4112\pm0.0005$, $4.2990\pm0.0008$, $0.09661\pm0.0005$, $0.9001\pm0.0012$ and $0.8797\pm0.0008$ respectively on the data set with corresponding 6 evaluation metrics. Our method still shows competitiveness, where the model remains fixed, the learning rate and the number of iterations are not changed, and the $\lambda_2$ of Equation 6 is adjusted to 0.25. (5) Our approach vs. DLDL (Age estimation (Gao et al. (2017))): Two age estimation datasets are used in our experiments. The first is Morph (Ricanek & Tesafaye (2006)), which is one of the largest publicly available age datasets. The second dataset is from the apparent age estimation competition, the first competition track of the ICCV ChaLearn LAP 2015 workshop (Escalera et al. (2015)). We employ the DPM model to detect the main facial region. Then, the detected face is fed into a cascaded convolution network to get the five facial key points, including the left/right eye centers, nose tip, and left/right mouth corners. Finally, based on these facial points, we align the face to the upright pose. The implementation details of our method with the preprocessing unit are referenced in DLDL. We used MAE and $\epsilon$-error to evaluate the performance of our model and the comparison method, respectively. DLDL algorithm on Morph and ChaLearn with metrics MAE ↓ and $\epsilon$-error yields performance that is {2.42, 0.23; 3.51, 0.31}. Our algorithm achieves competitive results (**{2.40, 0.20; 3.23, 0.27}**).

**Visualize of Label Distribution Matrix.** We conduct an experiment to verify the validity of our algorithm. We extract one sample (including features and labeled distributions) on Movie, and then

visualize the matrix of label distribution ($5 \times 10$) learned by the network and the corresponding label distribution. As shown in the Figure 3, we notice that the data distribution (**means and variances**) of the label distribution vector (each row of the label distribution matrix) is consistent with the corresponding label distribution.

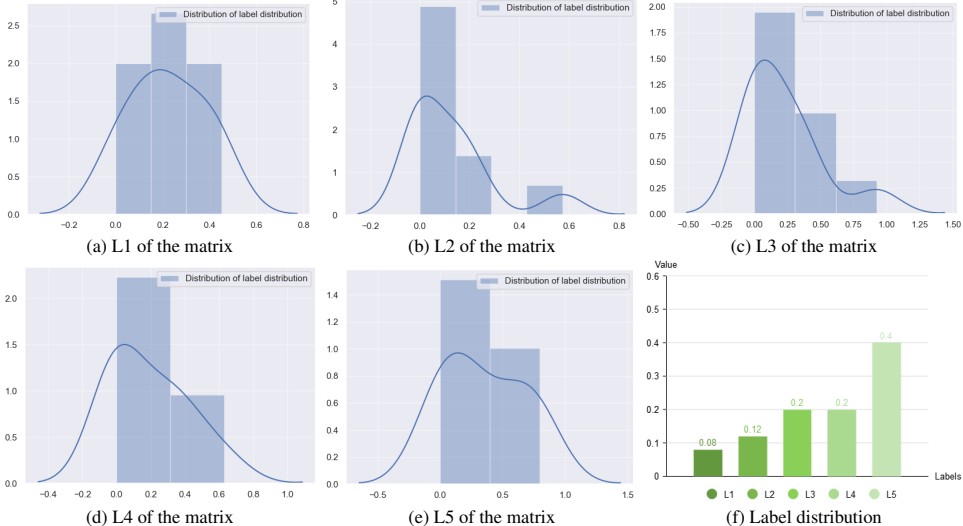

Figure 3: This figure shows the distribution of data for each vector of the label distribution matrix and the corresponding label for this label distribution matrix.

## 5 APPLICATION AND ENERGY CONSUMPTION

We select the algorithm executed on Gene as the evaluation model. The evaluation model is conducted on three platforms (lynxi HP300, Raspberry Pi, and Apple smartphones) to check the energy consumption with the same number of iterations (the power is evaluated thanks to the adb script (Dzhagaryan et al. (2016))). Specifically, our model is trained on Gene with the accuracy of float16. Then, the baseline model (MLP replaces SNN and GNN) and our model run inference on a mobile platform, each model executing 500 epochs with 32 batch sizes in the iteration. Experimental results show that our algorithm saves 34.6%, 41.2%, and 40.8% energy compared to the baseline algorithm on the three platforms, respectively.

## 6 RELATED WORKS

**Label distribution Learning.** Label distribution learning has attracted several attention as a new learning paradigm. Label distribution learning comes from the scheme proposed by (Geng (2016)) to address the age estimation task. Since then a large number of approaches have been proposed, such as low-rank hypothesis-based (Jia et al. (2019); Ren et al. (2019b)), metric-based (Gao et al. (2018)), manifold-based (Wang & Geng (2021)), and label correlation-based (Teng & Jia (2021); Qian et al. (2022)). Among them, some approaches are executed in computer vision (Chen et al. (2021a)), and speech recognition (Si et al. (2022)) tasks to improve the performance of classifiers. In this paper, we try to build a distribution of label distributions to moderate noise and uncertainty.

**Implicit Neural Representations.** In implicit neural representation, an object is usually represented as a multi-layer perception (MLP) that maps coordinates to a signal. This idea has been widely applied in modeling 3D object shapes (Lin et al. (2020); Kohli et al. (2020)), 3D surfaces of the scene (Sitzmann et al. (2019); Yariv et al. (2020); Niemeyer et al. (2020); Jiang et al. (2020)), the appearance of the 3D structure as well as the 2D image enhancement (Skorokhodov et al. (2021); Chen et al. (2021b); Anokhin et al. (2021); Karras et al. (2021)). In this paper, we seek to explore this technique to address the label distribution learning issue.

## 7 CONCLUSION

In this paper, we design an implicit distribution representation algorithm to moderate the uncertainty of the label values, where the implicit function can be a good estimate of the continuous distribution space. Furthermore, Gaussian prior methods and self-attention mechanisms help the model learn both local signals and global information of the label distribution matrix. Numerous experiments have verified the effectiveness of our approach as well as the suitability of the regularization technique. The application session demonstrates the high efficiency of our proposed model.

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

# 8 APPENDIX

The appendix is supplemented with the parameter settings of the network and the performance of all comparison algorithms on multiple data sets. Furthermore, we supplement the t-test in 12 data sets to validate the stability of our method. Since there are many hyperparameters, we conduct a parametric analysis on a Gene dataset.

| ID | Dataset | Examples | Features | Labels | Number of neurons of the SNN | LF | DA |
|----|---------|----------|----------|--------|------------------------------|-----|-----|
| 1 | wc-LDL | 500 | 243 | 12 | $\mathcal{L}_{1,2:end} \to [243, 1024]$ | Eq.5 | ✓ |
| 2 | SJAFFE | 213 | 243 | 6 | $\mathcal{L}_{1,2:end} \to [243, 1024]$ | Eq.5 | ✓ |
| 3 | SBU-3DFE | 2500 | 243 | 6 | $\mathcal{L}_{1,2:end} \to [243, 1024]$ | Eq.5 | ✗ |
| 4 | Scene | 2000 | 294 | 9 | $\mathcal{L}_{1,2:end} \to [294, 1024]$ | Eq.5 | ✗ |
| 5 | Gene | 17892 | 36 | 68 | $\mathcal{L}_{1,2:end} \to [36, 64]$ | Eq.6 | ✗ |
| 6 | Movie | 7755 | 1869 | 5 | $\mathcal{L}_{1,2:end} \to [1869, 1024]$ | Eq.5 | ✗ |
| 7 | M2B | 1240 | 250 | 5 | $\mathcal{L}_{1,2:end} \to [256, 1024]$ | Eq.5 | ✓ |
| 8 | SCUT | 1500 | 300 | 5 | $\mathcal{L}_{1,2:end} \to [300, 1024]$ | Eq.5 | ✓ |
| 9 | fbp5500 | 5500 | 512 | 5 | $\mathcal{L}_{1,2:end} \to [512, 1024]$ | Eq.5 | ✗ |
| 10 | RAF-ML | 4908 | 200 | 6 | $\mathcal{L}_{1,2:end} \to [200, 1024]$ | Eq.5 | ✗ |
| 11 | Twitter | 10040 | 200 | 8 | $\mathcal{L}_{1,2:end} \to [200, 1024]$ | Eq.5 | ✗ |
| 12 | Flickr | 11150 | 200 | 8 | $\mathcal{L}_{1,2:end} \to [200, 1024]$ | Eq.5 | ✗ |

Table 5: Statistics of the experimental datasets with models.

| ID | Dataset | Batch size | Epoch | Learning rate | Weight decay | Early stopping | Greed soup |
|----|---------|-----------|-------|---------------|--------------|----------------|------------|
| 1 | wc-LDL | 500 | 200 | $2 \times 10^{-3}$ | $1 \times 10^{-4}$ | ✓ | ✓ |
| 2 | SJAFFE | 213 | 200 | $2 \times 10^{-2}$ | $1 \times 10^{-4}$ | ✓ | ✓ |
| 3 | SBU-3DFE | 1000 | 200 | $1 \times 10^{-3}$ | $1 \times 10^{-4}$ | ✓ | ✓ |
| 4 | Scene | 1000 | 120 | $1 \times 10^{-3}$ | $1 \times 10^{-4}$ | ✓ | ✓ |
| 5 | Gene | 5000 | 150 | $2 \times 10^{-3}$ | $1 \times 10^{-4}$ | ✓ | ✓ |
| 6 | Movie | 2000 | 100 | $2 \times 10^{-3}$ | $1 \times 10^{-4}$ | ✓ | ✓ |
| 7 | M2B | 500 | 150 | $2 \times 10^{-3}$ | $1 \times 10^{-4}$ | ✓ | ✓ |
| 8 | SCUT | 500 | 150 | $1 \times 10^{-2}$ | $1 \times 10^{-4}$ | ✓ | ✓ |
| 9 | fbp5500 | 1500 | 300 | $2 \times 10^{-2}$ | $1 \times 10^{-4}$ | ✓ | ✓ |
| 10 | RAF-ML | 2000 | 100 | $1 \times 10^{-3}$ | $1 \times 10^{-4}$ | ✓ | ✓ |
| 11 | Twitter | 5000 | 200 | $1 \times 10^{-3}$ | $1 \times 10^{-4}$ | ✗ | ✗ |
| 12 | Flickr | 5000 | 200 | $1 \times 10^{-2}$ | $1 \times 10^{-4}$ | ✓ | ✓ |

Table 6: Training configuration of our model.

| Dataset | Algorithm | Chebyshev ↓ | Clark ↓ | Canberra ↓ | K-L ↓ | Cosine ↑ | Intersection ↑ |
|---------|-----------|-------------|---------|------------|-------|----------|----------------|
| RAF-ML | Ours | 0.1456 ± 0.0021 | 1.3651 ± 0.0441 | 2.6888 ± 0.0023 | 0.2017 ± 0.0012 | 0.9394 ± 0.0026 | 0.8247 ± 0.0077 |
| | a-LDL | 0.1466 ± 0.0032 | 1.3858 ± 0.0441 | 2.6999 ± 0.0063 | 0.2112 ± 0.0025 | 0.9300 ± 0.0031 | 0.8199 ± 0.0075 |
| | LDL-LRR | 0.1526 ± 0.0033 | 1.5651 ± 0.0111 | 2.7594 ± 0.0422 | 0.2449 ± 0.0007 | 0.9251 ± 0.0003 | 0.8141 ± 0.0044 |
| | LDL-LCLR | 0.1515 ± 0.0022 | 1.592 ± 0.0117 | 2.7779 ± 0.0239 | 0.2244 ± 0.0030 | 0.9262 ± 0.0062 | 0.8189 ± 0.0098 |
| | LDLSF | 0.1488 ± 0.0024 | 1.3889 ± 0.0086 | 2.7672 ± 0.0660 | 0.2302 ± 0.0044 | 0.9111 ± 0.0051 | 0.8117 ± 0.0022 |
| | LALOT | 0.1479 ± 0.0010 | 1.3659 ± 0.0099 | 2.6956 ± 0.0144 | 0.2221 ± 0.0064 | 0.9311 ± 0.0021 | 0.8107 ± 0.0008 |
| | BFGS-LLD | 0.1499 ± 0.0009 | 1.6656 ± 0.0066 | 2.7101 ± 0.0211 | 0.2541 ± 0.0055 | 0.9204 ± 0.0023 | 0.8157 ± 0.0050 |
| Twitter | Ours | 0.2777 ± 0.0021 | 2.2374 ± 0.0110 | 5.1163 ± 0.0018 | 0.5111 ± 0.0029 | 0.8807 ± 0.0049 | 0.7891 ± 0.0014 |
| | a-LDL | 0.2985 ± 0.0011 | 2.3002 ± 0.0112 | 5.4444 ± 0.0065 | 0.5242 ± 0.0033 | 0.8554 ± 0.0046 | 0.7709 ± 0.0084 |
| | LDL-LRR | 0.3129 ± 0.0021 | 3.2441 ± 0.0031 | 6.1454 ± 0.0023 | 0.6616 ± 0.0035 | 0.8002 ± 0.0042 | 0.7411 ± 0.0014 |
| | LDL-LCLR | 0.2994 ± 0.0045 | 2.4900 ± 0.0012 | 6.9609 ± 0.0041 | 0.6056 ± 0.0031 | 0.7110 ± 0.0021 | 0.7110 ± 0.0088 |
| | LDLSF | 0.3007 ± 0.0002 | 2.7887 ± 0.0057 | 5.6101 ± 0.0118 | 0.6396 ± 0.0022 | 0.7939 ± 0.0098 | 0.7660 ± 0.0007 |
| | LALOT | 0.3133 ± 0.0021 | 2.3141 ± 0.0016 | 5.5336 ± 0.0241 | 0.5233 ± 0.0021 | 0.8595 ± 0.055 | 0.7214 ± 0.0049 |
| | BFGS-LLD | 0.3114 ± 0.0044 | 2.5511 ± 0.0028 | 5.7145 ± 0.0041 | 0.5461 ± 0.0153 | 0.8335 ± 0.0055 | 0.7744 ± 0.0020 |
| Flickr | Ours | 0.2816 ± 0.0031 | 2.3356 ± 0.0097 | 5.2222 ± 0.0159 | 0.5314 ± 0.0033 | 0.8406 ± 0.0043 | 0.7741 ± 0.0025 |
| | a-LDL | 0.2998 ± 0.0088 | 2.4388 ± 0.0089 | 5.3111 ± 0.0119 | 0.5729 ± 0.0067 | 0.8331 ± 0.0009 | 0.7500 ± 0.0033 |
| | LDL-LRR | 0.3329 ± 0.0012 | 3.4400 ± 0.0174 | 6.3459 ± 0.0229 | 0.6516 ± 0.0031 | 0.8450 ± 0.0040 | 0.7399 ± 0.0037 |
| | LDL-LCLR | 0.2970 ± 0.0009 | 2.4444 ± 0.0063 | 6.1600 ± 0.0041 | 0.6222 ± 0.0013 | 0.7919 ± 0.0029 | 0.7090 ± 0.0070 |
| | LDLSF | 0.3301 ± 0.0009 | 2.8888 ± 0.0459 | 5.9152 ± 0.0121 | 0.6100 ± 0.0021 | 0.8139 ± 0.0098 | 0.7360 ± 0.0037 |
| | LALOT | 0.3411 ± 0.0026 | 2.9140 ± 0.0019 | 5.3333 ± 0.0243 | 0.5737 ± 0.0012 | 0.8225 ± 0.020 | 0.7144 ± 0.0004 |
| | BFGS-LLD | 0.3200 ± 0.0041 | 2.7517 ± 0.0060 | 5.8149 ± 0.0048 | 0.5961 ± 0.0099 | 0.8131 ± 0.0011 | 0.7407 ± 0.0077 |

Table 7: The performance of our proposed method with the comparison algorithms on 12 datasets.

**t-test on 12 datasets.** We evaluate the range of p-values for the six metrics on 12 data sets. Cheby.$[1.65e - 105, 1.00e + 00]$, Clark$[6.87e - 97, 1.99e - 02]$, Canbe.$[9.75e - 99, 1.23e - 01]$, KL$[1.99e - 101, 1.23e - 01]$, Cosine$[1.02e - 98, 2.99e - 01]$, and Inter.$[5.03e - 112, 3.67e - 01]$ According to the test results, the LDL methods have significantly different performance in terms of each metric on all datasets except Gene (at a 0.05 significance level). The label distribution of Gene tends to be uniformly distributed, which may result in the equal performance of the LDL approaches.

**Parameter Sensitivity Analysis.** Our method has three parameters, including the regularization parameter $\lambda_1$, $\lambda_2$, and $\beta$. To analyze the sensitivity of $\lambda_1$, $\lambda_2$, and $\beta$, we run our method with three sets ($\{0.001, 0.005, 0.01, 0.05, 0.1\}$, $\{0.001, 0.005, 0.01, 0.05, 0.1\}$, and $\{0.001, 0.005, 0.08, 0.02, 0.08\}$) on the Gene dataset (see Figure 4).

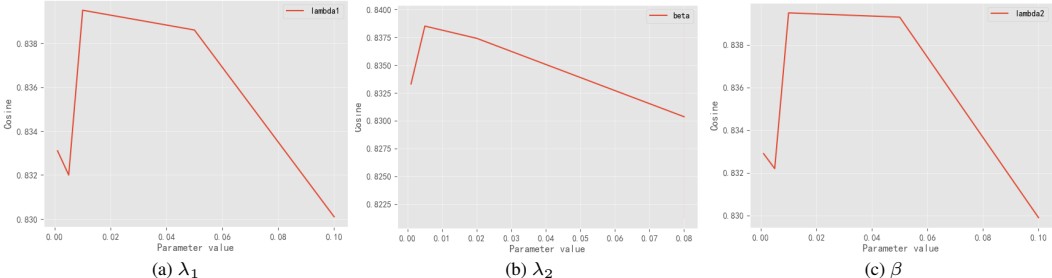

(a) $\lambda_1$        (b) $\lambda_2$        (c) $\beta$

Figure 4: This figure shows the sensitivity of parameters on the Gene dataset.

**Formula Supplement.**

(1) Scaled Dot-Product attention:

$$Attention(Q, K, V) = \text{softmax}(\frac{QK^T}{\sqrt{d_k}})V,$$

where $\sqrt{d_k}$ is the dimension of the key vector $k$ and query vector $q$ .

(2) ReLU:

$$\text{ReLU}(z) = max(0, z).$$

(3) L1 loss:

$$\sum_{i=1}^{D} |x_i - y_i|.$$

(4) KL divergence:

$$\text{KL}(\hat{y}||y) = \sum_{c=1}^{M} \hat{y}_c \log \frac{\hat{y}_c}{y_c}.$$

(5) Perceptual loss:

$$L1(\text{MLP}(x), \text{MLP}(y)).$$

