# OpenReview forum: "Label Distribution Learning via Implicit Distribution Representation"
_ICLR.cc/2023/Conference — Submitted to ICLR 2023_

### Official Review · Reviewer_JLrG · 2022-10-19

**Confidence:** 4
**Correctness:** 4
**Technical Novelty And Significance:** 3
**Empirical Novelty And Significance:** 3
**Recommendation:** 8

**Clarity, Quality, Novelty And Reproducibility:**

Clarity: The paper is well-organized and clearly written.

Quality: Technically solid paper, with a high impact on the research field of label distribution learning.

Novelty: The paper makes non-trivial advances over the current state-of-the-art.

Reproducibility:  Key details are sufficiently well-described for competent researchers to confidently reproduce the main results.

**Strength And Weaknesses:**

### Strength

- In my opinion, the motivation for this paper is interesting. In detail, both noise and uncertainty in label distribution learning are important and urgent problems to be solved. In addition, the introduction of deep implicit representation learning is also a very valuable attempt.
- This paper provides some novel perspectives. First, the introduction of implicit label distribution representation brings an effective way to characterize and mitigate the uncertainty of the label distribution. Second,  the proposed Label Distribution Matrix Learning also presents a novel view for estimating the uncertainty of the labels. Besides, the regularization techniques designed in this paper can give some inspiration to researchers in the field of label distribution learning.
- For the empirical studies, this paper conducts extensive experiments and demonstrates the advantages of the proposed method over the state-of-the-art LDL methods, which can effectively reflect the performance of the algorithm.
- The paper is well-organized and clearly written, which is easy to follow.

### Weaknesses

- I am concerned about the self-attention mechanism in the method. Concretely, the authors mention that, "to capture the global correlation between labels to generate a standard label distribution, we employ a self-attention mechanism to model the label distribution matrix". However, in my opinion, this statement is not strongly supported. I suggest the authors further analyze how the self-attention mechanism captures the global correlation between labels.
- Although the proposed method seems to be very effective compared with existing methods, some statistical methods should be used to illustrate the experimental results, such as t-test and Bonferroni-Dunn test, to confirm this observation.
- I find there are many hyper-parameters that should be tuned in the experiments, such as batch size, the number of epochs, learning rate, the weights $\lambda_1$, $\lambda_2$ and $\beta$. The authors should analyze the influence of these parameters on the final performances.



**Summary Of The Paper:**

This paper introduces the implicit distribution in the label distribution learning framework to handle the uncertainty of each label value in the label distribution training set. It uses deep implicit representation learning to construct a label distribution matrix with Gaussian prior constraints to moderate the noise and uncertainty interference of the label distribution dataset, and label distributions are transformed by using the self-attention algorithm through each row component of the label distribution matrix.

The main contributions of this paper are:

- It points out that the label distribution dataset used for training has a high probability of inaccuracy and uncertainty, which significantly limits the performance of LDL algorithms.
- This paper introduces the implicit distribution in the label distribution learning framework to characterize the uncertainty of each label value.
- This paper adopts Spiking neural network with an MLP to save energy consumption of mobile devices, and correlations between labels are deeply mined by a graph convolutional network.
- This paper designs some regularization techniques to boost the performance of the model and a new LDL dataset is released.

**Summary Of The Review:**

If the authors can address my concerns well, I would tend to accept this paper.

---

> ### Author Response · Authors · 2022-11-09
> **We sincerely thank the reviewer for his/her constructive comments.**
>
> Dear reviewer JLrG,
> thanks a lot for reviewing our paper and giving us valuable suggestions.
>
> Q1：I am concerned about the self-attention mechanism in the method. Concretely, the authors mention that "to capture the global correlation between labels to generate a standard label distribution, we employ a self-attention mechanism to model the label distribution matrix". However, in my opinion, this statement is not strongly supported. I suggest the authors further analyze how the self-attention mechanism captures the global correlation between labels.
>
> A1: As a suggestion, we add a description related to the self-attention mechanism in the revised version, and in addition, we visualize the gains from the method in the form of a heat map (see Figure 2).
>
> Q2：Although the proposed method seems to be very effective compared with existing methods, some statistical methods should be used to illustrate the experimental results, such as the t-test and Bonferroni-Dunn test, to confirm this observation.
>
> A2: We add this experiment to the appendix of the revised version to check the effectiveness of our method, and the experiment confirms that our method is reliable. In addition, we find that the general performance of the Gene dataset may be due to its tending to have a uniform distribution in the label space.
>
> Q3：I find should any hyper-parameters should be tuned in the experiments, such as batch size, the number of epochs, learning rate, and the weights  λ1, λ2 and β. The authors should analyze the influence of these parameters on the final performances.
>
> A3: As a suggestion, we add this part of the experiments in the appendix section of the revised manuscript, and we only evaluate the cosine metric for the time being because of the high training cost of the deep model.

---

### Official Review · Reviewer_eABN · 2022-10-21

**Confidence:** 3
**Clarity, Quality, Novelty And Reproducibility:** The writing can be improved for easie…
**Correctness:** 3
**Technical Novelty And Significance:** 2
**Empirical Novelty And Significance:** 3
**Recommendation:** 5

**Strength And Weaknesses:**

Strengths:
1. The paper is to the best of my knowledge the first to try out the improvement of SNN on LDL.
2.  This paper conducts sufficient experiments, and the results look superior.

Weaknesses:
1. The proposed method is not well-motivated. This paper provides a dense description of the algorithm, but it is difficult for the reader to understand the ideas behind the method. I think it is interesting idea to use SNNs to solve LDL, but it requires more reasoning about why it has the potential to go beyond existing methods and what is its unique methodological contribution given quite a number of existing SNN works.
2. The paper lacks formal introductions of the problem studied (LDL) and several techniques used (SNN, GCN, et al.). It is hard to understand basic concepts without reading the reference, so it would be better to explain more details about the background.
3. The baselines use different model and some of them just use a linear model. So I am concerned whether the superiority of the proposed algorithm comes from the complex network structure.

**Summary Of The Paper:**

The paper proposes an implementation of the spike-based learning method for the label distribution learning (LDL) problem. Main idea is to use a spiking neural network (SNN) to construct a latent feature space in which the coordinate matrix learned from a graph convolution network look up the table. The experimental results showed that the proposed method can achieve high accuracy in several tasks compared to existing methods.

**Summary Of The Review:**

The paper presents a nice step in an interesting direction, but it does not clarify what exactly its innovations are.

---

> ### Author Response · Authors · 2022-11-09
> **We sincerely thank the reviewer for his/her constructive comments.**
>
> Dear reviewer eABN,
> thanks a lot for reviewing our paper and giving us valuable suggestions.
>
> Q1: The proposed method is not well-motivated. This paper provides a dense description of the algorithm, but it is difficult for the reader to understand the ideas behind the method. I think it is an interesting idea to use SNNs to solve LDL, but it requires more reasoning about why it has the potential to go beyond existing methods and what is its unique methodological contribution given quite several existing SNN works.
>
> A1: First of all, for the label distribution learning task, the label value is first inferred from a coarse-grained discernment of the existence of the label in the scene, followed by a fine-grained evaluation of the description of the label. Our network follows this inference pattern by first using SNN for coarse-grained logical decisions and then using MLP for accurate regression.
> SNNs meet this requirement and are more efficient than conventional deep models. Furthermore, compared to the existing work on SNNs, we are the first to propose a strategy using a mixture of SNNs and MLPs, aiming to combine the advantages of SNNs and ANNs to learn LDL tasks.
>
> Q2: The paper lacks formal introductions of the problem studied (LDL) and several techniques used (SNN, GCN, et al.). It is hard to understand basic concepts without reading the reference, so it would be better to explain more details about the background.
>
> A2: As a suggestion, we add background and motivation to further explain our work in the revised manuscript.
>
> Q3： The baselines use a different model and some of them just use a linear model. So I am concerned about whether the superiority of the proposed algorithm comes from the complex network structure.
>
> A3: As a suggestion, in the revised version, we add a baseline model to validate the superiority of our proposed representation technique.
>
> For example, we set up the a-LDL baseline algorithm on the wc-LDL dataset as shown in the table below.
>
> | Algorithm  | Chebyshev  | Clark   | Canberra  | K-L  | Cosine  | Intersection  |
> | :----: | :----: | :----: |  :----: |  :----: |  :----: |  :----: |
> | Ours   | 0.0779 | 0.3980 | 0.7779| 0.4040 | 0.9883 | 0.8778 |
> | a-LDL | 0.0855 | 0.4667 | 0.8007| 0.4455 | 0.9788 | 0.8705 |
> | LDL-LRR| 0.1122 | 0.4772 | 0.8802| 0.5533 | 0.9510 | 0.8555 |
> | LDL-LCLR| 0.1057 | 1.0569 | 0.7890| 0.5045 | 0.9668 | 0.8383 |
> | LALOT| 0.0989| 0.6689 | 0.8089| 0.4778 | 0.9476 | 0.8700 |
> | BFGS-LDL| 0.1229 | 1.5657 | 0.7998| 0.4998 | 0.9704 | 0.8611 |

---

> > ### Comment · Reviewer_eABN · 2022-11-28
> > **Thanks for the response**
> >
> > I appreciated the additional explanations addressed most of my concerns.
> >
> > Let me say that I am not an expert on the relevant techniques (e.g. SNN) used, but I am relatively familiar with LDL, the subject of study in the paper. I think this work is an interesting attempt in the direction of LDL, but the proposed method seems to be domain specific. As LDL a general paradigm, but can it only be applied to visual tasks? The paper also uses mixup, but no ablation experiments were done to demonstrate where the advantage of the algorithm comes from.

---

> > > ### Author Response · Authors · 2022-11-28
> > > **Many thanks to the reviewers for their responses.**
> > >
> > > Dear reviewer eABN, thanks a lot for reviewing our paper and giving us valuable suggestions.
> > >
> > > Q1: Is it a generic algorithm?
> > >
> > > A1:  First of all, it is important to state that our approach is a general strategy, and only the operators of the latent feature extractor need to be changed for text, image, and audio data.
> > >
> > > Q2: Did the experiment verify the effectiveness of the method?
> > >
> > > A2: The reviewer (7uiR) also raises relevant concerns, and we add extensive ablation experiments in response to the Reviewer's (7uiR) mini-post to demonstrate the effectiveness of our approach.
> > >
> > >
> > > The following are additional ablation experiments.
> > >
> > >
> > > Q： Demonstrating the role of implicit neural representations.
> > >
> > > **Key messages:**
> > > This makes me uneasy, because it makes it seem like if you just use a $\color{red}{comparable} $ &nbsp; $\color{red}{model}$ &nbsp; $\color{red}{size}$ for the a-LDL baseline, you would get comparable results to the proposed method. I don't know whether nor not this is true.
> > >
> > > As a suggestion, we set up two new models (b-LDL, c-LDL) and their sizes are almost the same size as our method. In Section 3 (**latent feature extraction**), both b-LDL and c-LDL have 17-layer networks with the same activation functions as ours, where b-LDL uses all SN (spiking neural) layers and c-LDL uses the standard MLP. Note that the number of neurons is set to match that of Table 5.
> > >
> > > The headache is the GCN section. In Section 3 (**learning label distribution matrix**), our GCN includes four graph convolution layers and four activation layers. GCN is parallel to the sub-network (**latent feature extraction**). In our ablation experiments, b-LDL and c-LDL are connected in series with 4 MLPs and 4 activation layers after sub-network, and finally, the self-attention module is kept after **learning label distribution matrix**. b-LDL and c-LDL use regularization terms and data augmentation techniques that are consistent with our method. This new ablation experiment is conducted on the wc-LDL dataset. The c-LDL has challenged our approach in a small number of metrics.
> > >
> > > | Algorithm  | Chebyshev  | Clark   | Canberra  | K-L  | Cosine  | Intersection  |
> > > | :----: | :----: | :----: |  :----: |  :----: |  :----: |  :----: |
> > > | Ours   | 0.0779 | 0.3980 | 0.7779| 0.4040 | 0.9883 | 0.8778 |
> > > | w/o perceptual loss (ours)   | 0.0781 | 0.3988 | 0.7778| 0.4041 | 0.9883 | 0.8777 |
> > > | w/o perceptual loss (a-LDL)  | 0.0859 | 0.4665 | 0.8012| 0.4452 | 0.9789 | 0.8700 |
> > > | b-LDL  | 0.0792 | 0.4002 | 0.7899| 0.4130 | 0.9880 | 0.8771 |
> > > | c-LDL  | 0.0782 | 0.3979 | 0.7786| 0.4053 | 0.9883 | 0.8772|
> > > | a-LDL with AD | 0.0844 | 0.4652 | 0.8001| 0.4413 | 0.9799 | 0.8763 |
> > > | a-LDL | 0.0855 | 0.4667 | 0.8007| 0.4455 | 0.9788 | 0.8705 |
> > > | LDL-LRR| 0.1122 | 0.4772 | 0.8802| 0.5533 | 0.9510 | 0.8555 |
> > > | LDL-LCLR| 0.1057 | 1.0569 | 0.7890| 0.5045 | 0.9668 | 0.8383 |
> > > | LALOT| 0.0989| 0.6689 | 0.8089| 0.4778 | 0.9476 | 0.8700 |
> > > | BFGS-LDL| 0.1229 | 1.5657 | 0.7998| 0.4998 | 0.9704 | 0.8611 |
> > >
> > > Analysis.
> > > Implicit representation learning focuses on correlation constraints between labels compared to other comparison methods. The excellent performance of the LDL-LRR algorithm verifies that the label correlation constraint is useful. In addition, graph networks by nature capture deep semantics better than label rank loss.

---

> ### Author Response · Authors · 2022-11-14
> **We sincerely thank the reviewer for his/her constructive comments.**
>
> Dear reviewer eABN,
>
> Thanks a lot for reviewing our paper and giving us valuable suggestions.
>
> We have tried our best to answer all the questions according to the comments. We sincerely hope that our responses could address all your concerns. Is there anything that needs us to further clarify for the given concerns?
>
> Thanks again for your hard work.

---

### Official Review · Reviewer_vTyL · 2022-10-24

**Confidence:** 3
**Correctness:** 3
**Technical Novelty And Significance:** 3
**Empirical Novelty And Significance:** 3
**Recommendation:** 6

**Clarity, Quality, Novelty And Reproducibility:**

Some details are missing, such as the definition of \mathcal{L}_{kl}. To make the paper easy to follow, it is better for the paper to be reorganized, such as the application of the augmentations in the loss function. There exist some novelties in the proposed method, such as the regularization technique.


**Strength And Weaknesses:**

Strength:
1.	The idea of considering the label distribution as an implicit neural representation is interesting.
2.	The experimental results show that the proposed method is promising.
3.	In the proposed method, a novel augmentation strategy is proposed to improve the performance of LDL.

Weakness:
1.	Some details of the proposed method are missing, such as the definition of \mathcal{L}{{kl} and the representation of the augmentation samples in the function.
2.	In the proposed method, a novel augmentation strategy is proposed with mask. In the ablation study, the effectiveness of the proposed strategy has not been verified by comparing with mixup and so on. The main contributions of the proposed method should be further verified in the experiments.
3.	Meanwhile, more details about the proposed method should be presented, such as how the implicit distribution characterize the uncertainty of each label value and how the model mitigrate the uncertainty of the label distribution.


**Summary Of The Paper:**

Since the complexity of the manual annotation task or the inaccuracy of the label enhancement algorithm leads to noise and uncertainty in the label distribution training set, it is hard to generate accurate label distribution. This paper proposes a novel framework for label distribution learning by introducing implicit neural representation. It is achieved by generating a coordinate matrix, and then using this matrix to sample latent features, which is extracted by a deep spiking neural network. In addition, a series of regularization techniques are used to mitigate the overfitting problem. The experiments have proven the effectiveness of the proposed method.

**Summary Of The Review:**

This paper proposes a novel framework for label distribution learning by introducing implicit neural representation. It is achieved by generating a coordinate matrix, and then using this matrix to sample latent features, which is extracted by a deep spiking neural network. However, some details of the proposed strategy are missing, and the written of the paper should be further improved.

---

> ### Author Response · Authors · 2022-11-09
> **We sincerely thank the reviewer for his/her constructive comments.**
>
> Dear reviewer vTyL,
> thanks a lot for reviewing our paper and giving us valuable suggestions.
>
> Q1: Some details of the proposed method are missing, such as the definition of \mathcal{L}{{kl} and the representation of the augmentation samples in the function.
>
> A1: As a suggestion, we check the method description section and add related content in the revised manuscript.
>
> Q2: In the proposed method, a novel augmentation strategy is proposed with a mask. In the ablation study, the effectiveness of the proposed strategy has not been verified by comparing it with mixup and so on. The main contributions of the proposed method should be further verified in the experiments.
>
> A2: In the revised version, we add this ablation experiment and the evaluation results show that our method outperforms the standard mixup by an average of 6 percent. In addition, we add a baseline model to demonstrate the effectiveness of the method.
>
> Q3: Meanwhile, more details about the proposed method should be presented, such as how the implicit distribution characterizes the uncertainty of each label value and how the model mitigates the uncertainty of the label distribution.
>
> A3: As a suggestion, we add to the revised manuscript by providing detailed descriptions and experiments.
>
> 1) The latent features look up the table in the coordinate system to build a label distribution matrix.
>
> 2) The label distribution matrix is constrained by a regularization term that comes from the result of a Gaussian sampling of the label space.
>
> Specifically, assuming a label distribution with a value of 0.2, we use a Gaussian function to extend it, with the mean of the Gaussian function set to 0.2 and the variance to 0.2, for which 2L points are sampled as a vector.
> In addition, random sampling inevitably generates noise in the build-up of the label distribution matrix, so the self-attention module learns from the uncertainty representation to avoid noise disturbance.

---

> ### Author Response · Authors · 2022-11-14
> **We sincerely thank the reviewer for his/her constructive comments.**
>
> Dear reviewer vTyL,
>
> Thanks a lot for reviewing our paper and giving us valuable suggestions.
>
> We have tried our best to answer all the questions according to the comments. We sincerely hope that our responses could address all your concerns. Is there anything that needs us to further clarify for the given concerns?
>
> Thanks again for your hard work.

---

### Official Review · Reviewer_7uiR · 2022-11-04

**Confidence:** 3
**Correctness:** 2
**Technical Novelty And Significance:** 3
**Empirical Novelty And Significance:** 2
**Recommendation:** 5

**Clarity, Quality, Novelty And Reproducibility:**

Quality  and clarity of writing is below average. In particular, the notation used in the Method section is confusing and ambiguous. I encourage the authors to proofread the Methods section.

The model architecture is novel. However, the authors do not evaluate the architecture by itself, so it is hard to determine whether the performance gains seen in Table 3 can be attributed to the novel architecture or the additional regularization tricks being used.

**Strength And Weaknesses:**

Strengths:
(1) The empirical results appear strong and definitely exceed state-of-the-art.

Weaknesses:
(1) The only major argument I see against acceptance would be lack of novelty. I will split comments into 2 parts: (i) innovations in neural architecture (model/backbone design) and (ii) optimization objective (loss function).

(i) The architecture takes as input data features and outputs a discrete distribution over possible labels. The authors make two innovations: (a) using SNN and (b) training a label distribution matrix, which is an estimate of the distribution over label distributions. (b) is their main contribution, since it is highlighted in the title of the work. (a) is a minor contribution, since it is an off-the-shelf component used in a new setting.

Regarding the label distribution matrix:

Q1: From the paper, it is not clear *how novel* the label distribution matrix (implicit distribution representation) part of the work is? Specifically, looking at the paragraph with the heading ``Label Distribution Matrix Learning'', I see numerous citations regarding confidence and uncertainty learning. How is this work different?

Q2: From my reading of the paper, I do not see a robust justification for the addition of this label distribution matrix. Why do we expect this addition to the model architecture to improve results?

Q3: Why is the shape of the label distribution matrix L x L x 2 and the distribution values 1 x 2L? Specifically, where does the 2 come from?

(ii) I find the optimization objective to be a collection of standard approaches and therefore not novel (All the content after "Regularization Techniques" in Section 2). Specifically:

(a) The linear normalization function is just the standard way of normalizing a vector of values with a constant offset.

(b) Data augmentation: Mix-up is one of the most ubiquitous tricks in computer vision. Random masking is similarly ubiquitous (Drop-out, for example).

(c) The loss function consists of L1 and KL distances, regularizing a distribution towards a Gaussian, perceptual loss ... these are all standard.

Experimental Results:

Table 3 shows that the entire method as a package achieves state-of-the-art on many benchmarks. However, given that the novel part of the method is the model *architecture*, I would expect some architecture-to-architecture comparisons. For example, "Ours" in the Table includes mix-up, random masking, and some regularization; do the baselines use these techniques? If not, I would consider these comparisons unfair, since these techniques are straightforward to apply.

The first two rows and Table 4 show that on the Gene dataset, removing the perceptual loss from the proposed framework lowers the performance to be about the same as the state-of-the-art. This is concerning, since this suggests that all of the gains are coming from the perceptual loss.

Other comments:

Q4: Why does the Gaussian regularization term make sense? (referring to the last term in equation (6) where you regularize the matrix $\mathcal{M}$ towards the constant $\overline{\mathcal{M}}$) If I understand correctly, each row of matrix $\mathcal{M}_i$ is a distribution over probability of the true label being label $i$. This is a number between 0 and 1. (see Figure 2). A distribution over the range [0,1] cannot be a Gaussian. For example, Figure 2(e) shows that the model gave some weight to the probability of true label being L5 to be over 1.0 or under 0.0. The true label cannot be L5 with 150% probability. Furthermore, the label distribution has to add up to one -- that is, every time you sample a column vector using the rows of $\mathcal{M}$ as the probability distribution, you should get $L$ non-negative values that sum up to 1. So it doesn't make sense to me why you would regularize each row of $\mathcal{M}$ to be an independent Gaussian. Perhaps I misunderstood somethings -- please clarify.

Minor comments:
(1) Notation in the method section is ambiguous and hard to follow:

(a) C is used in Equation (2); it has already been used to denote the coordinate matrix, perhaps use $K$?

(b) Before Equation (2), you use bold-case $\mathbf{d}_i$ to denote a vector and $d^y_i$ to denote scalar entries into this vector. Later on in equation (5) and (6), you do not bold the vectors $\hat{d}_i$ and $d_i$. Adding to the confusion, you use both superscripts and subscripts to index vectors and matrices, so the math is hard to follow.

(c) Eq. 5: Please be specific as to which norm is used in $\|\| \cdot \|\|$. Please specify which distributions are being compared by the KL loss term $\mathcal{L}_{kl}$

(d) Eq. 5: in the last term index $i$ is abused. The way you wrote it is looks as though the loss only operates on the diagonal of matrix $\mathcal{M}$.

(e) In the sentence after Eq. (5), `` $d_i$ is the label distribution ... $\hat{d}$ is the ground truth''. Is this a typo? Please do not use hats to indicate ground truth.

(f) Eq. (4) Please do not use $\times$ for element wise vector multiplication, it is confusing.

(g) In Eq. 4 the $y_i$s are clearly vectors. In equation (2), you use the exact same symbol as scalar indices into the vector $\mathbf{d}_i$ as $d_i^{y_i}$.

(h) Eq. (1) Please either use the operator $\exp d$ or write $e^d$.

(i) Tables: Bolding the standard deviation does not make sense. I recommend bolding the best number instead.

(j) There is no Appendix to this paper. While that by itself is not an issue, there are small experimental details throughout the paper that would logically belong in an Appendix. For example, in the Method section, the authors spend many sentences describing details such as " each linear unit contains 1024 neurons ... contains 64 neurons ... we attempted other activation functions ReLU, Swish, Sigmoid ...''. There are details regarding the specific library and version used to implement the method -- all before the actual method is presented.

**Summary Of The Paper:**

This paper describes a method for label distribution learning (LDL). By my understanding, the contributions of this work fall into two categories: (i) network architecture and (ii) optimization objective.
For (i), the authors use an SNN (Spiking neural network) and a Graph Convolutional Neural Network (GCN) to generate a label distribution matrix. The label distribution matrix is the "implicit distribution representation" referred to in the title and abstract. Intuitively, this matrix is a distribution over label distributions, which are regularized towards Gaussians.
For (ii), the authors use L1 , KL, and perceptual losses, along with mix-up, random masking and linear normalization to yield their final framework.

**Summary Of The Review:**

This paper appears marginal. At first glance, there is some novelty with regards to the model architecture (specifically, all the stuff between input features and output label distribution), and there are some convincing empirical results.

However, most empirical results presented are based on a combination of the novel architecture changes and several standard regularization tricks. Consequently, most comparisons are in my opinion unfair, and the empirical significance of the architecture contributions is unclear.

---

> ### Author Response · Authors · 2022-11-09
> **We sincerely thank the reviewer for his/her constructive comments**
>
> Dear reviewer 7uiR,
> thanks a lot for reviewing our paper and giving us valuable suggestions.
>
> Q: Innovation.
>
> A: We restate the contribution and motivation in the revised manuscript (see Section 2), which are summarized as the following 3 points.
>
> 1. Existing LDL algorithms focus only on how to model the features-to-labels mapping on a customized dataset. In contrast, our work explores the uncertainty of the label distribution space itself with the help of implicit representation learning on a customized dataset.
>
> 2. We are the first to propose to characterize the uncertainty of the labels in the representation space of the model by using a label distribution matrix.
>
> 3. In contrast to existing LDL methods that enforce a regularization term on the model via a pre-defined correlation of labels (R1-R4), we learn the latent relationships of such labels by using GCN and self-attention.
>
> [R1] Facial emotion distribution learning by exploiting low-rank label correlations locally, CVPR 2019.
>
> [R2] Label distribution learning with label correlations via low-rank approximation, IJCAI 2019.
>
> [R3] Label distribution learning by exploiting label distribution manifold, TNNLS 2021.
>
> [R4] A label distribution manifold learning algorithm, PR 2022.
>
>
> Network Architecture.
>
> Q1: From the paper, it is not clear how novel the label distribution matrix (implicit distribution representation) part of the work is?  Specifically, looking at the paragraph with the heading ``Label Distribution Matrix Learning'', I see numerous citations regarding confidence and uncertainty learning. How is this work different?
>
> A1: Existing approaches (R5-R8) to uncertainty modeling employ a label distribution learning paradigm to enforce loss terms in the training phase to boost the robustness of the model (classifier). Unlike these approaches, our study focuses on the uncertainty of the label distribution itself and alleviates the impact of label uncertainty via uncertainty representation (label distribution matrix).
>
> [R5] Towards unbiased label distribution learning for facial pose estimation using anisotropic spherical gaussian, ECCV 2022.
>
> [R6] Unimodal-Concentrated Loss: Fully adaptive label distribution learning for ordinal regression, CVPR 2022.
>
> [R7] Age estimation using the expectation of label distribution learning, IJCAI 2018.
>
> [R8] Sense beauty by label distribution learning, IJCAI 2017.
>
> Q2: From my reading of the paper, I do not see a robust justification for the addition of this label distribution matrix. Why do we expect this addition to the model architecture to improve results?
>
> A2: From a deep modeling perspective, the goal is twofold: 1. Interpretability, we attempt to make the representations of deep networks meaningful, which in turn helps us understand the process of learning uncertainty modeling from label distributions; 2. Controllability, the label distribution matrix can be directly executed by attention mechanisms, MLP, convolution, and other operators to learn the required feature maps. From the meaning of the label distribution matrix itself, we aim to characterize the distribution of the label distribution to depict its uncertainty.
>
> Q3: Why is the shape of the label distribution matrix L x L x 2 and the distribution values 1 x 2L? Specifically, where do the 2 come from?
>
> A3: The dimension from the label distribution matrix is L x 2L. In the revised manuscript, we add the explanation that the number 2 is mainly from the assumption that the label relationship is modeled in a plane coordinate system, and we also attempt to model in higher dimensions but 2 is the best in terms of execution efficiency.
>
> Regularization method.
>
> The regularization method is not our main work, and we provide in the revised version a different architecture algorithm with the label distribution matrix to verify the effectiveness of our method. The experimental results show that this baseline model is slightly weaker than our algorithm and stronger than the other baselines.
>
> Experimental Results.
>
> Our comparison is fair and ``Ours'' includes regularization techniques; moreover, the gain in perceptual loss is not significant from the data, as we verify in the revised manuscript.
>
> Q4： Why does the Gaussian regularization term make sense?
>
> A4:    Literature [R8] demonstrates that the Gaussian distribution is the best compared to other distribution functions in the LDL task. As a suggestion, we adopt the reviewer's comment to bound the output of Gaussian distribution in the range of 0 to 1 in the revised manuscript.
>
> Minor comments: We thank the reviewers for their comments, and we check each of them in the revised manuscript.

---

> > ### Comment · Reviewer_7uiR · 2022-11-18
> > **Response to rebuttal**
> >
> > Thank you, authors, for the detailed rebuttal and the revision. You addressed most of my concerns. I still have concerns regarding the experimental results.
> >
> > Since the main contribution of the paper is the implicit representation, ideally there would be strong results showing that performance gains in Table 1 (Ours compared to a-LDL) comes from only the implicit representation and not the perceptual loss (or mixup).
> >
> > From the rebuttal: authors assert that "the gain in perceptual loss is not significant from the data, as we verify in the revised manuscript."
> >
> > I'm somewhat skeptical of this assertion.
> >
> > In the manuscript in Section 4 page 5, authors explain that "For the baseline model (a-LDL), we drop the implicit representation and use only a two-layer network with the label distribution matrix to learn a label distribution. a-LDL without any additional regularization terms and data augmentation mechanisms in the training stage. " I assume this means that a-LDL does not use perceptual loss, please correct me if I'm wrong.
> >
> > If this is true, then none of the results in Table 1 show an advantage of the implicit representation.
> >
> > In Table 2, there is one result on the Gene dataset where the authors remove the perceptual loss. This result suggests that all the gains presented in Table 1 are coming from the perceptual loss. Specifically, without the perceptual loss, on the Gene dataset, the proposed method obtains slightly worse results than a-LDL in terms of intersection, slightly worse in terms of Cosine, slightly worse in terms of K-L, significantly worse in terms of Canberra, and slightly better on the other two metrics (but well within a standard deviation).
> >
> > Furthermore, the proposed method used better data augmentation than the a-LDL baseline.
> >
> > In summary, in my opinion, there is *not enough results supporting the claim that the implicit representation improves performance*.

---

> > > ### Author Response · Authors · 2022-11-19
> > > **We sincerely thank the reviewer for his/her constructive comments.**
> > >
> > > Dear reviewer 7uiR, thanks a lot for reviewing our paper and giving us valuable suggestions.
> > >
> > > Q: Does perception loss play an important role? (The role of implicit neural representation.)
> > >
> > > A: First of all, a-LDL with perceptual loss is enforced on the training dataset, which we will add to the manuscript. Second, we conduct a new ablation experiment in wc-LDL without perceptual loss. As shown in the table below, our approach without perceptual loss does not improve the performance significantly, and even some metrics have dropped.
> > >
> > > | Algorithm  | Chebyshev  | Clark   | Canberra  | K-L  | Cosine  | Intersection  |
> > > | :----: | :----: | :----: |  :----: |  :----: |  :----: |  :----: |
> > > | Ours   | 0.0779 | 0.3980 | 0.7779| 0.4040 | 0.9883 | 0.8778 |
> > > | w/o perceptual loss (ours)   | 0.0781 | 0.3988 | 0.7778| 0.4041 | 0.9883 | 0.8777 |
> > > | w/o perceptual loss (a-LDL)  | 0.0859 | 0.4665 | 0.8012| 0.4452 | 0.9789 | 0.8700 |
> > > | a-LDL with AD | 0.0844 | 0.4652 | 0.8001| 0.4413 | 0.9799 | 0.8763 |
> > > | a-LDL | 0.0855 | 0.4667 | 0.8007| 0.4455 | 0.9788 | 0.8705 |
> > > | LDL-LRR| 0.1122 | 0.4772 | 0.8802| 0.5533 | 0.9510 | 0.8555 |
> > > | LDL-LCLR| 0.1057 | 1.0569 | 0.7890| 0.5045 | 0.9668 | 0.8383 |
> > > | LALOT| 0.0989| 0.6689 | 0.8089| 0.4778 | 0.9476 | 0.8700 |
> > > | BFGS-LDL| 0.1229 | 1.5657 | 0.7998| 0.4998 | 0.9704 | 0.8611 |
> > >
> > > Analysis.
> > >
> > > Perceptual loss plays an important role when our algorithm is run on a dataset with a large label space (one instance corresponding to 68 labels in Gene dataset). The contribution of perceptual loss is not significant on the wc-LDL dataset (12 labels). Notably, here we conduct a parameter sensitivity analysis for a-LDL fine-tuning.
> > >
> > > New ablation experiments are added in the revised manuscript, which verifies that perceptual loss only acts significantly in large label spaces. I hope this answer solves the reviewers' doubts and obtains a better rating for our work.

---

> > > > ### Comment · Reviewer_7uiR · 2022-11-20
> > > > **Response**
> > > >
> > > > Thank you for this.
> > > >
> > > > I agree that on the wc-LDL the perceptual loss is not needed.
> > > >
> > > > I don't think I understand correct exactly what a-LDL is. I saw this sentence: "a-LDL without any additional regularization terms and data augmentation mechanisms in the training stage. ". I understood this to mean no perceptual loss.
> > > >
> > > > I also saw this sentence: "a-LDL performs sub-optimally probably because the depth of the model is insufficient." This makes me uneasy, because it makes it seem like if you just use a comparable model size for the a-LDL baseline, you would get comparable results to the proposed method. I don't know whether nor not this is true.
> > > >
> > > > Looking at Table 2, the text indicates there is supposed to be an ablation experiment (e) using mixup. But I don't see Table 2 (e) in the latest revision. Am I looking at the right table?
> > > >
> > > > There is a clear performance gain of "ours" over "a-LDL". But there are a few differences, not just the novel implicit representation with Gaussian priors. Therefore, I don't think this claim is justified: "our approach has a huge performance gain with the help of implicit distribution representation with Gaussian priors." There is no ablation result that clearly shows the performance gain from only the implicit representation.

---

> > > > > ### Author Response · Authors · 2022-11-20
> > > > > **Response about the role of our implicit neural representation.**
> > > > >
> > > > > Dear reviewer 7uiR,
> > > > > thanks to this critical suggestion for we to verify the role of implicit neural representations.
> > > > >
> > > > > Q：Demonstrating the role of implicit neural representations.
> > > > >
> > > > > **Key messages:**
> > > > > This makes me uneasy, because it makes it seem like if you just use a $\color{red}{comparable} $ &nbsp; $\color{red}{model}$ &nbsp; $\color{red}{size}$ for the a-LDL baseline, you would get comparable results to the proposed method. I don't know whether nor not this is true.
> > > > >
> > > > > As a suggestion, we set up two new models (b-LDL, c-LDL) and their sizes are almost the same size as our method. In Section 3 (**latent feature extraction**), both b-LDL and c-LDL have 17-layer networks with the same activation functions as ours, where b-LDL uses all SN (spiking neural) layers and c-LDL uses the standard MLP. Note that the number of neurons is set to match that of Table 5.
> > > > >
> > > > > The headache is the GCN section. In Section 3 (**learning label distribution matrix**), our GCN includes four graph convolution layers and four activation layers. GCN is parallel to the sub-network (**latent feature extraction**). In our ablation experiments, b-LDL and c-LDL are connected in series with 4 MLPs and 4 activation layers after sub-network, and finally, the self-attention module is kept after **learning label distribution matrix**. b-LDL and c-LDL use regularization terms and data augmentation techniques that are consistent with our method. This new ablation experiment is conducted on the wc-LDL dataset. The c-LDL has challenged our approach in a small number of metrics.
> > > > >
> > > > > | Algorithm  | Chebyshev  | Clark   | Canberra  | K-L  | Cosine  | Intersection  |
> > > > > | :----: | :----: | :----: |  :----: |  :----: |  :----: |  :----: |
> > > > > | Ours   | 0.0779 | 0.3980 | 0.7779| 0.4040 | 0.9883 | 0.8778 |
> > > > > | w/o perceptual loss (ours)   | 0.0781 | 0.3988 | 0.7778| 0.4041 | 0.9883 | 0.8777 |
> > > > > | w/o perceptual loss (a-LDL)  | 0.0859 | 0.4665 | 0.8012| 0.4452 | 0.9789 | 0.8700 |
> > > > > | b-LDL  | 0.0792 | 0.4002 | 0.7899| 0.4130 | 0.9880 | 0.8771 |
> > > > > | c-LDL  | 0.0782 | 0.3979 | 0.7786| 0.4053 | 0.9883 | 0.8772|
> > > > > | a-LDL with AD | 0.0844 | 0.4652 | 0.8001| 0.4413 | 0.9799 | 0.8763 |
> > > > > | a-LDL | 0.0855 | 0.4667 | 0.8007| 0.4455 | 0.9788 | 0.8705 |
> > > > > | LDL-LRR| 0.1122 | 0.4772 | 0.8802| 0.5533 | 0.9510 | 0.8555 |
> > > > > | LDL-LCLR| 0.1057 | 1.0569 | 0.7890| 0.5045 | 0.9668 | 0.8383 |
> > > > > | LALOT| 0.0989| 0.6689 | 0.8089| 0.4778 | 0.9476 | 0.8700 |
> > > > > | BFGS-LDL| 0.1229 | 1.5657 | 0.7998| 0.4998 | 0.9704 | 0.8611 |
> > > > >
> > > > > Analysis.
> > > > > Implicit representation learning focuses on correlation constraints between labels compared to other comparison methods. The excellent performance of the LDL-LRR algorithm verifies that the label correlation constraint is useful. In addition, graph networks by nature capture deep semantics better than label rank loss.
> > > > >
> > > > > Experiment (e) of Table 2 will be added to this section in a subsequent revised manuscript.

---

> > > > > ### Author Response · Authors · 2022-11-27
> > > > > **We sincerely thank the reviewer for his/her constructive comments.**
> > > > >
> > > > > Dear reviewer 7uiR,
> > > > > thanks a lot for reviewing our paper and giving us valuable suggestions.
> > > > >
> > > > > I hope this supplemented experiment addresses your concerns, do you have any other questions?

---

> ### Author Response · Authors · 2022-11-14
> **We sincerely thank the reviewer for his/her constructive comments.**
>
> Dear reviewer 7uiR,
>
> Thanks a lot for reviewing our paper and giving us valuable suggestions.
>
> We have tried our best to answer all the questions according to the comments. We sincerely hope that our responses could address all your concerns. Is there anything that needs us to further clarify for the given concerns?
>
> Thanks again for your hard work.

---

### Official Review · Reviewer_Paus · 2022-11-12

**Confidence:** 3
**Correctness:** 3
**Technical Novelty And Significance:** 3
**Empirical Novelty And Significance:** 2
**Recommendation:** 5

**Clarity, Quality, Novelty And Reproducibility:**

The overall presentation of this paper is mostly ok, but still hard to follow in a few places. It lacks a concrete example to motivate the paper, and gives a very intuitive introduction to the considered contexts. Although the paper works on a relatively new and important problem, it is not clear what is the technical novelty of this paper, since each part of the proposed workflow has been extensively studied before. In addition, one concern is that the paper mostly discusses what the proposed methods are, but lacks a discussion on intuition and insights, e.g., why each component is designed in the proposed way, and why not other alternatives. These could be clearly justified.

**Strength And Weaknesses:**

Strength:

- Important and challenging problem
- Sound and feasible solution
- Extensive evaluation to demonstrate the potential usefulness
- Promising results

Weakness:

- The presentation needs to be improved, lacking motivation examples, discussion on intuition and insights.
- Unclear about the technical novelty, which is more an assemble of existing techniques.
- Unclear why energy consumption could be a concern of this paper, since there are quite a big bunch of community that works on model compression and optimization for diverse edge devices
- Unclear how to extend to streaming data cases.

**Summary Of The Paper:**

The polysemy of data introduces a big challenge for the labeling process. This paper proposes an implicit distribution representation method based on label distribution learning, which also considers the uncertainty of label values, in which the Gaussian prior and self-attention-based methods are also adopted for learning the local and global label distribution matrix. A relatively large-scale evaluation on 12 datasets is performed, confirming the potential of the proposed methods.

**Summary Of The Review:**

Overall, this paper proposes a sound and feasible solution for label distribution learning. The evaluation results also show to be promising. However, it still posts a few concerns, regarding technical novelty, contribution, and application to a more wide range of applications. This paper works on an important problem, and these concerns should be carefully discussed and justified for further enhancement.

---

> ### Author Response · Authors · 2022-11-12
> **We sincerely thank the reviewer for his/her constructive comments.**
>
> Dear reviewer Paus,
> thanks a lot for reviewing our paper and giving us valuable suggestions.
>
> Q: Innovation.
>
> A: We restate the contribution and motivation in the revised manuscript (see Section 2), which are summarized as the following 3 points.
>
> 1. Existing LDL algorithms focus only on how to model the features-to-labels mapping on a customized dataset. In contrast, our work explores the uncertainty of the label distribution space itself with the help of implicit representation learning on a customized dataset.
> 2. We are the first to propose to characterize the uncertainty of the labels in the representation space of the model by using a label distribution matrix.
> 3. In contrast to existing LDL methods that enforce a regularization term on the model via a pre-defined correlation of labels (R1-R4), we learn the latent relationships of such labels by using GCN and self-attention.
>
> [R1] Facial emotion distribution learning by exploiting low-rank label correlations locally, CVPR 2019.
>
> [R2] Label distribution learning with label correlations via low-rank approximation, IJCAI 2019.
>
> [R3] Label distribution learning by exploiting label distribution manifold, TNNLS 2021.
>
> [R4] A label distribution manifold learning algorithm, PR 2022.
>
> Q: Motivation and insight for framework design.
>
> 1)	Why consider using implicit representation learning?
>
> In contrast to existing approaches that consider 2D and 3D, we attempt to build a generative model by decoding the 1D space, where the label relations can be considered as spatial relations on 1D coordinates. In addition, the implicit neural representation is more powerful in representation ability compared to the existing LDL algorithm, which is well-suited to build an implicit function to learn continuous regression tasks (R5).
>
> [R5] Continuous label distribution learning, PR 2022.
>
> 2)	SNN.
>
> The label value is first inferred from a coarse-grained discernment of the existence of the label in the scene, followed by a fine-grained evaluation of the description of the label. Our network follows this inference pattern by first using SNN for coarse-grained logical decisions and then using MLP for accurate regression.
> SNNs meet this requirement and are more efficient than conventional deep models. Furthermore, compared to the existing work on SNNs, we are the first to propose a strategy using a mixture of SNNs and MLPs, aiming to combine the advantages of SNNs and ANNs to learn LDL tasks.
>
>
> 3)	Label distribution matrix.
>
> Existing approaches (R6-R9) to uncertainty modeling employ a label distribution learning paradigm to enforce loss terms in the training phase to boost the robustness of the model (classifier). Unlike these approaches, our study focuses on the uncertainty of the label distribution itself and alleviates the impact of label uncertainty via uncertainty representation (label distribution matrix).
> From a deep modeling perspective, the goal is twofold: 1. Interpretability, we attempt to make the representations of deep networks meaningful, which in turn helps us understand the process of learning uncertainty modeling from label distributions; 2. Controllability, the label distribution matrix can be directly executed by attention mechanisms, MLP, convolution, and other operators to learn the required feature maps. From the meaning of the label distribution matrix itself, we aim to characterize the distribution of the label distribution to depict its uncertainty.
>
> [R6] Towards unbiased label distribution learning for facial pose estimation using anisotropic spherical gaussian, ECCV 2022.
>
> [R7] Unimodal-Concentrated Loss: Fully adaptive label distribution learning for ordinal regression, CVPR 2022.
>
> [R8] Age estimation using the expectation of label distribution learning, IJCAI 2018.
>
> [R9] Sense beauty by label distribution learning, IJCAI 2017.
>
> As a suggestion, we add related motivations and insights in the revised manuscript.

---

> ### Author Response · Authors · 2022-11-14
> **We sincerely thank the reviewer for his/her constructive comments.**
>
> Dear reviewer Paus,
>
> Thanks a lot for reviewing our paper and giving us valuable suggestions.
>
> We have tried our best to answer all the questions according to the comments. We sincerely hope that our responses could address all your concerns. Is there anything that needs us to further clarify for the given concerns?
>
> Thanks again for your hard work.

---

### Decision · Program_Chairs · 2023-01-20

**Decision:**

Reject

**Justification For Why Not Higher Score:**

With three of the five reviewers not willing to advocate for accepting the paper, it does not appear that the paper is ready for publication.  The second reviewer noted several issues with the paper in particular.

**Justification For Why Not Lower Score:**

N/A

**Metareview: Summary, Strengths And Weaknesses:**

Thanks for your submission to ICLR.

This paper ended up after discussion with three somewhat negative reviews and two somewhat to reasonably positive reviews.  In terms of strengths, the reviewers noted that it was an interesting problem tackled and there are some promising results.  On the negative side, multiple reviewers noted issues in presentation / details in the paper, multiple reviewers took issue with the novelty, and multiple reviewers noted some issues with the experimental results.

After discussion, some of these issues seemed to be resolved but others were still unresolved.  After discussion, three of the reviewers still remained on the negative side, not advocating for accepting the paper.  Unfortunately, it seems that there's not enough here to warrant accepting the paper for ICLR.  It seems with some additional work this paper could be quite strong.  Please do keep in mind the suggestions of the reviewers when preparing a future version of the manuscript.